# Causal Bayesian Optimization with Unknown Causal Graphs

## Abstract

Causal Bayesian Optimization (CBO) is a methodology designed to optimize an outcome variable by leveraging known causal relationships through targeted interventions. Traditional CBO methods require a fully and accurately specified causal graph, which is a limitation in many real-world scenarios where such graphs are unknown. To address this, we propose a new method for the CBO framework that operates without prior knowledge of the causal graph. We demonstrate through theoretical analysis and empirical validation that focusing on the direct causal parents of the target variable is sufficient for optimization. Our method learns a Bayesian posterior over the direct parents of the target variable. This allows us to optimize the outcome variable while simultaneously learning the causal structure. Our contributions include a derivation of the closed-form posterior distribution for the linear case. In the nonlinear case where the posterior is not tractable, we present a Gaussian Process (GP) approximation that still enables CBO by inferring the parents of the outcome variable. The proposed method performs competitively with existing benchmarks and scales well to larger graphs, making it a practical tool for real-world applications where causal information is incomplete.

## 1 Introduction

Many real-world applications require optimizing the outcome of a function that is both unknown and expensive to evaluate. For instance, in agriculture, this optimization focuses on determining which fertilizers to apply in order to maximize crop yield. In robotics, it involves fine-tuning control algorithms to achieve optimal performance. In medical applications, such as in drug discovery, the objective is to maximize the efficacy of a drug. *Bayesian Optimization* (BO) (Jones et al., 1998) is a widely used framework used to solve these kinds of optimization problems. The framework uses a surrogate model to estimate uncertainty in the unknown function and strategically optimizes it through a series of well-chosen queries.

Typically, BO focuses on a black-box setup. As a result, BO methods have no notion of a targeted intervention and thus requires intervening on all variables at each iteration of the algorithm in order to optimize the function of interest. This is limiting as interventions are costly. *Causal Bayesian Optimization* (CBO) (Aglietti et al., 2020) leverages causal information in the form of a causal graph (Pearl, 2009) to provide more structure to this black-box function. In a causal graph, the arrows point from *cause* to *effect*, which offers insights into the relationship between the variables. CBO uses this causal information to select *targeted interventions that optimize an outcome variable of interest*. As a motivating example, CBO can be used for a doctor to prescribe drugs to minimize the *prostate-specific antigen* (PSA) levels of a patient. By incorporating causal knowledge, we can make more informed decisions about which interventions are likely to improve the outcome. This means we can optimize the outcome variable with targeted interventions. Leveraging causal information can therefore reduce the dimensionality of the BO problem.

Although the CBO methodology has been extended to various settings, such as model-based approaches (Sussex et al., 2023; 2024), functional interventions (Gultchin et al., 2023), dynamic settings (Aglietti et al., 2021), and constrained settings (Aglietti et al., 2023), a fundamental limitation of these approaches is the assumption of a known causal graph. In practice, the causal graph is almost never fully known, and even domain experts may not correctly identify all causal relationships. This limitation motivates the setting where we consider the CBO problem in the case where

the causal graph is unknown. While Branchini et al. (2023) address this by proposing an acquisition function that jointly learns the full causal graph and optimizes the target variable, it does not scale to larger unknown graphs. We demonstrate through theoretical analysis and empirical validation that focusing exclusively on the direct causal parents of the target variable is sufficient for optimization. This enables the CBO framework to scale to more complex graphs.

In this paper, we propose a method for optimizing the CBO objective with *an unknown causal graph*. The objective of our approach is two-fold: we need to optimize the CBO objective and simultaneously learn the causal graph. Specifically, we introduce a Bayesian posterior over the direct causal parents of the target variable. We focus on using the interventional data to learn the direct causal parents of the target variable rather than the full graph. These are the variables that directly influence the target variable. In the healthcare scenario for instance, the doctor would want to prescribe the drugs that directly influence the PSA levels. As we perform more interventions, the method iteratively improves both our understanding of the CBO objective and the direct causal parents of the target variable. The iterative procedure of the methodology is given in Figure 1.

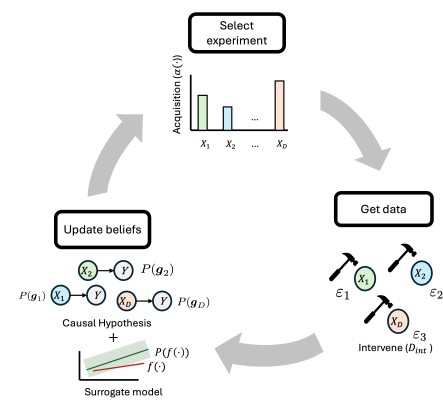

Figure 1: An overview of the iterative process of the method where the interventions are used to update beliefs about the surrogate model and the direct parents.

Our main contributions are

1. We derive a posterior distribution that learns the direct parents of the target variable using both observational and interventional data, and incorporate this posterior into the surrogate model.

2. We theoretically establish that specifying only the direct parents is sufficient for optimization under some commonly made assumptions and empirically show that it converges to the same optimal value as when the full graph is used.

3. We demonstrate that our method achieves the same optimal target value on synthetic and semi-synthetic causal graphs as methods with fully specified causal graphs.

4. We provide more general benchmarks for CBO where we show that our method scales to graphs of up to at least 100 nodes.

## 2 RELATED WORK

**Causal Bayesian Optimization.** There are various methods that build on the CBO methodology from Aglietti et al. (2020). Aglietti et al. (2021) consider the case where both the target variable and the input variables evolve over time. Furthermore, Sussex et al. (2023; 2024) propose model-based CBO methods that have cumulative regret guarantee. Aglietti et al. (2023) explore the constrained case where a constraint condition is added to CBO. Gultchin et al. (2023) consider a functional intervention. In contrast to these, our work considers the CBO problem in scenarios where *the causal graph is unknown*. Branchini et al. (2023) address the same problem, but introduce an acquisition function to jointly optimize the target and learn the structure; however, their approach fails to scale to larger graphs. In contrast, our work presents a scalable framework specifically designed for solving the CBO problem with completely unknown causal graphs. Unlike (Branchini et al., 2023), we focus on identifying the direct causal parents of the target variable. We also consider the case where hard interventions are performed.

**Causal Discovery.** There has been a recent line of work for Bayesian inference over causal graphs using continuous optimization (Annadani et al., 2021; Lorch et al., 2021; Cundy et al., 2021; Hägele et al., 2023; Annadani et al., 2024), but learning the full graph is not needed in our optimization problem. Causal Discovery methods that combine observational and interventional data is known as joint causal inference (Mooij et al., 2020). There are various methods that learn the full graph

using joint causal inference (Hauser & Bühlmann, 2012; Magliacane et al., 2016; Wang et al., 2017; Yang et al., 2018), but all these methods learn a point estimate of the graph. Local Causal Discovery centers on identifying the direct causal parents of a target variable. One such approach is to use invariant prediction to learn parents of a target variable (Schölkopf et al., 2012; Peters et al., 2016; Ghassami et al., 2017; Heinze-Deml et al., 2018). Invariant prediction is based on the idea that conditional distribution of the target variable will not change if we intervene on all the parents of that variable. Another approach is to use a combination of double machine learning and iterative feature selection to infer the direct causal parents of a target variable from observational data (Soleymani et al., 2020; Angelis et al., 2023; Quinzan et al., 2023). However, these methods give point estimates for the causal structure. We are interested in a posterior distribution of the direct parents. This captures the epistemic uncertainty over the direct causal parents.

**Causal Bandits.** Since the introduction of causal bandits by Lattimore et al. (2016), there has been a line of work to use causal bandits for optimal decision making. Specifically, Lee & Bareinboim (2018) introduce optimalilty conditions for hard interventions in the bandit setting. There has been work done on causal bandits with unknown graphs (Lu et al., 2021; Malek et al., 2023; Yan & Tajer, 2024), but the methods are restricted to specific graph types, additive functional relations or linear bandits. In this work, we consider the CBO objective with hard interventions and nonlinear functional relationships between the variables.

## 3 PRELIMINARIES

**Structural Causal Models and Interventions.** In this work, we leverage the framework of probabilistic *Structural Causal Models* (SCMs) to capture the causal relationships between the random variables. SCMs use *Directed Acyclic Graphs* (DAGs) to model causal relations, where the arrows in the graph point from *causes* to *effects*. Consider the DAG $\mathcal{G}$ and a tuple $\langle \boldsymbol{U}, \boldsymbol{V}, \boldsymbol{F}, p(\mathbf{U}) \rangle$, where: $\boldsymbol{U} = \{u_1, u_2, \ldots, u_K\}$ is the set of exogenous (or unobserved) noise variables, and $p(\boldsymbol{U})$ is the corresponding distribution. $\boldsymbol{V} = \{v_1, v_2, \ldots, v_K\}$ is the set of endogenous (or observed) variables. $\boldsymbol{F} = \{f_1, f_2, \ldots, f_K\}$ is a set of causal mechanisms that relate the exogenous and endogenous variables such that $v_k = f_k(\mathbf{pa}_k, u_k) \quad \forall k = 1, 2, \ldots, K$, where $\mathbf{pa}_k$ denotes the set of parents of variable $v_k$ in the DAG $\mathcal{G}$. A DAG $\mathcal{G}$ that satisfies the Causal Markov Condition is said to be Markovian. The Causal Markov Condition says that each endogenous variable is independent of its non-descendants given its direct causes (parents) in the DAG. For a Markovian DAG, the joint distribution over the endogenous variables can be factorized as $P(v_1, v_2, \ldots, v_K) = \prod_{k=1}^{K} P(v_k \mid \mathbf{pa}_k)$. This factorization is known as the observational distribution. When we intervene on a variable $X$ and set it to a specific value $x$, all the edges pointing into $X$ are removed from the original DAG, resulting in a modified submodel denoted as $\mathcal{G}^{\mathrm{do}(X=x)}$. This type of intervention is known as a *hard intervention*, and is written as $\mathrm{do}(X = x)$. Using this framework, the interventional distribution after an intervention on a variable $V_i$ can be written as

$$P(v_1, v_2, \ldots, v_K \mid \mathrm{do}(V_i = v_i)) = \prod_{k=1}^{K} P(v_k \mid \mathbf{pa}_k^{\mathrm{do}(V_i=v_i)}) \tag{1}$$

where $\mathbf{pa}_k^{\mathrm{do}(V_i=v_i)}$ is the parents of $V_k$ in the subgraph obtained after intervening on $V_i$. This expression highlights how interventions can change the dependencies in a causal model

**Causal Discovery.** This is the problem of learning the causal structure from the data. Both observational data ($\mathcal{D}_{\mathrm{obs}}$) and interventional data ($\mathcal{D}_{\mathrm{int}}$) can be used to learn the causal graph. If only observational data is available, the causal DAG can only be estimated up to the *Markov Equivalence Class* (MEC) (Spirtes et al., 2001; Peters et al., 2017). We can overcome the problem of only identifying the MEC by using interventional data as well. It is shown that using interventional data allows us to get a finer partioning of the DAG, which improves the identifiability of the causal models (Hauser & Bühlmann, 2012). In the Bayesian approach to causal discovery, rather than inferring a single DAG, we estimate the posterior distribution over the SCM (Friedman & Koller, 2003; Heckerman et al., 2006). Suppose $\mathcal{G}$ is the current graph we are considering and $\boldsymbol{\Theta}$ is the parameters of the SCM, then we can use Bayes rule to define the posterior of the parameters given to current data as $P(\mathcal{G}, \boldsymbol{\Theta} \mid \mathcal{D}) = P(\mathcal{D}, \boldsymbol{\Theta}, \mathcal{G})/P(\mathcal{D})$, where $P(\mathcal{D}, \boldsymbol{\Theta}, \mathcal{G}) = P(\mathcal{D} \mid \mathcal{G}, \boldsymbol{\Theta})P(\mathcal{G}, \boldsymbol{\Theta})$ are the likelihood and the prior respectively and $P(\mathcal{D}) = \sum_{\mathcal{G}} \int P(\mathcal{D} \mid \mathcal{G}, \boldsymbol{\Theta})P(\mathcal{G}, \boldsymbol{\Theta})\mathrm{d}\boldsymbol{\Theta}$ is the marginal.

This posterior is however intractable due to the superexponential growth of possible DAGs as the number of nodes increase (Robinson, 1977).

**Causal Bayesian Optimization.** BO is a method that minimizes an unknown black-box function $f : \mathbb{R}^d \to \mathbb{R}$ over a feasible set by using a surrogate model $P(f(\cdot))$ and an acquisition function $\alpha(\cdot)$ to iteratively select query points. The acquisition function is optimized to choose the next point, which is evaluated by the black-box function. After this the surrogate model is updated. CBO follows a framework similar to BO but focuses on selecting targeted interventions to optimize $Y$. Let $\mathcal{G}$ represent the causal DAG, where the causal mechanisms $\boldsymbol{F}$ are unknown. The target variable is $Y$, and the endogenous variables $\boldsymbol{V}$ are divided into manipulative variables $\boldsymbol{X}$ and non-manipulative variables $\boldsymbol{C}$. The intervention $\boldsymbol{\xi} := \boldsymbol{s}, \boldsymbol{v} = \mathrm{do}(\boldsymbol{X}_s = \boldsymbol{v})$ allows for interventions on more than one variable. The interventional distribution of the target variable is $P(Y \mid \boldsymbol{\xi}, \boldsymbol{C})$, and the observational distribution is $P(Y \mid \boldsymbol{X}s = \boldsymbol{v}, \boldsymbol{C})$. The class of Causal Global Optimization (CGO) problems is then defined as

$$\boldsymbol{\xi}^* = \underset{\boldsymbol{\xi} \in \mathcal{P}(\boldsymbol{X}), \mathcal{D}(\boldsymbol{X})}{\arg\min} \mathbb{E}_{P(Y|\boldsymbol{\xi},\boldsymbol{C})}[Y \mid \boldsymbol{\xi}, \boldsymbol{C}, \mathcal{G}] \tag{2}$$

where $\mathcal{P}(\boldsymbol{X})$ is the exploration set (ES) of $\boldsymbol{X}$ and $\mathcal{D}(\boldsymbol{X})$ is the corresponding interventional domain for each element in the ES. For example, if we consider the PSA example, then $\mathcal{P}(\boldsymbol{X})$ would represent the combinations of different medications that the doctor can prescribe to the patient and $\mathcal{D}(\boldsymbol{X})$ would be the corresponding dosages that the doctor can prescribe. In the standard problem $\mathcal{G}$ is a known DAG (Aglietti et al., 2020), but the case we are considering is that $\mathcal{G}$ is unknown (Branchini et al., 2023). The objective is to select the intervention that minimizes the expected value of the target variable $Y$. The *Gaussian Process* (GP) surrogate model for a specific intervention $\boldsymbol{s}$ is then defined as

$$f_{\boldsymbol{s}}(\boldsymbol{v}) \sim \mathcal{GP}(m(\boldsymbol{v}), K_C(\boldsymbol{v}, \boldsymbol{v}')) \tag{3}$$

where the mean function is $m(\boldsymbol{v}) = \hat{\mathbb{E}}[Y \mid \boldsymbol{\xi}, \boldsymbol{C}]$, the kernel function is $K_C(\boldsymbol{v}, \boldsymbol{v}') = K_{RBF}(\boldsymbol{v}, \boldsymbol{v}') + \sigma(\boldsymbol{v})\sigma(\boldsymbol{v}')$ and $K_{\mathrm{RBF}}(\cdot, \cdot)$ is the radial basis kernel function. Furthermore $m(\boldsymbol{v})$ and $\sigma(\boldsymbol{v})$ is estimated using the observational data and *do-calculus*.

# 4 METHODOLOGY

In this section, we discuss our solution to the CBO with unknown causal graphs (CBO-U), where we focus on learning the relevant edges in the problem. We are specifically considering the CBO problem in the case of hard interventions. Thus we are solving the problem for the cases where hard interventions are optimal.

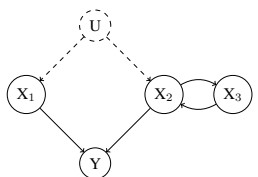

It has been shown that hard interventions are optimal when there are no spouses between $Y$ and the rest of the input features (Gultchin et al., 2023). Two variables are consider spouses if they are connected by a bidirected edge (or if they share a common child). The set of variables that are a spouse for $Y$ is written as $\mathbf{sp}_Y$. When such spouses are absent, directly intervening on the parents of $Y$ is sufficient for optimization because there are no hidden pathways or indirect effects influencing $Y$ through other variables. However,

Figure 2: Causal graph showing assumptions between variables for optimality of hard interventions.

intervening on the direct parents can become suboptimal when there exists variables that can act as context for functional interventions, especially if $Y$ is influenced by a confounder (Gultchin et al., 2023). In these cases, simply manipulating the parents of $Y$ may not capture the full effect on $Y$ because confounders can introduce additional dependencies or biases that affect the outcome. Therefore, we restrict ourselves to cases where such conditions do not apply, ensuring that hard interventions remain optimal. Furthermore, we make these assumptions as this mimics the intended setup for CBO, where the aim is to select interventions that optimize a downstream variable of interest. For example, in healthcare it would be the outcome of the patient and in agriculture it would be the crop yield. These target variables often do not have edges pointing to any of the input features.

Thus, in this work, we make the following assumptions:

**Assumption 4.1.** *$Y$ has no causal effect on any of the features.*

**Assumption 4.2.** *We can intervene on the direct parents of $Y$.*

**Assumption 4.3.** $Y$ *is not influenced by confounders.*

For the rest of the variables, we do not assume causal sufficiency. In the input features we can deal with both unobserved confounders and cycles. Appendix B.4 empirically shows that it is not always necessary to able to intervene on all the parents, but that it is still important to identify the non-manipulative parents for the algorithm to converge. The next question we need to address is whether intervening on these parent variables is sufficient for solving the CGO problem. To answer this, we refer to the following results from Lee & Bareinboim (2018) and Gultchin et al. (2023). Lee & Bareinboim (2018) show that the optimal action for $Y$ is to intervene on the direct causes of $Y$, provided that $Y$ is not confounded by unobserved confounders. Furthermore Gultchin et al. (2023) show that if $\mathbf{sp}_Y = \emptyset$ then the optimal intervention set are $\boldsymbol{X} \in \mathbf{pa}_Y$ with hard interventions as the optimal interventions.

### 4.1 ESTIMATING THE POSTERIOR DISTRIBUTION OF DIRECT PARENTS

In this work, we will use a special case of the SCM, namely the Gaussian Additive Noise Model (ANM). In the CBO problem we are interested in causal mechanism of the target variable $Y$. This can be written as

$$Y := f(\mathbf{pa}_Y) + \varepsilon_Y \text{ where } \varepsilon_Y \sim \mathcal{N}(0, \sigma_Y^2) \tag{4}$$

The Gaussian ANM is identifiable if the functions are not linear or constant in its arguments (Hoyer et al., 2008; Peters et al., 2014). We will use this ANM to derive a posterior distribution for a given set of parent variables. Let $\boldsymbol{g} \in \mathbb{R}^d$ be a vector where $[\boldsymbol{g}]_j = 1$ denotes that variable $j$ is a parent of $Y$ and $\boldsymbol{\theta}_Y$ be to parameters of $f(\cdot)$. Suppose we have a new point $\{\boldsymbol{x}, y\}$, then we want to determine the following probability

$$P(\boldsymbol{g}, \boldsymbol{\theta}_Y \mid \boldsymbol{x}, y) = \frac{P(\boldsymbol{g}, \boldsymbol{\theta}_Y, \boldsymbol{x}, y)}{P(\boldsymbol{x}, y)}. \tag{5}$$

Using Bayes theorem, the numerator can be written as $P(\boldsymbol{g}, \boldsymbol{\theta}_Y, \boldsymbol{x}, y) = P(\boldsymbol{x}, y \mid \boldsymbol{g}, \boldsymbol{\theta}_Y)P(\boldsymbol{\theta}_Y \mid \boldsymbol{g})P(\boldsymbol{g})$ and the denominator is $P(\boldsymbol{x}, y) = \sum_{\boldsymbol{g}} \int_{\boldsymbol{\theta}_Y} P(\boldsymbol{g}, \boldsymbol{\theta}_Y, \boldsymbol{x}, y) \mathrm{d}\boldsymbol{\theta}_Y$ where we sum over all possible parent sets. Since, we are using the ANM, the likelihood can be written as

$$P(\boldsymbol{x}, y \mid \boldsymbol{g}, \boldsymbol{\theta}_Y) = P_{\varepsilon_Y}(y - f(\boldsymbol{x}_{\mathrm{pa}_Y})) \equiv \mathcal{N}(y - f(\boldsymbol{x}_{\mathrm{pa}_Y}); 0, \sigma_Y^2) \tag{6}$$

**Linear Case.** We will first derive the posterior in closed form for the linear SCM. Suppose we have the following setting. Suppose that $\mathbf{pa}_Y = \boldsymbol{X}_s$, $\mathcal{D}_{\mathrm{obs}} = \{\boldsymbol{X}, \boldsymbol{y}\}$ where $\boldsymbol{X} \in \mathbb{R}^{d \times N}$, $\boldsymbol{y} \in \mathbb{R}^N$, $p = |\boldsymbol{X}_s|$ and we observe a sample $\{\boldsymbol{x}, y\}$, and we use the linear SCM which means $Y := \boldsymbol{\theta}_s^\top \boldsymbol{X}_s + \varepsilon_Y$ where $\varepsilon_Y \sim \mathcal{N}(0, \sigma_Y^2)$. The likelihood can be written as $P_{\varepsilon_Y}(y - \boldsymbol{\theta}_s^\top \boldsymbol{x}_s)$. Then we can determine the probability distribution for the parameters of the linear SCM as follows

$$P(\boldsymbol{\theta}_s \mid \boldsymbol{g}_s, \boldsymbol{X}, \boldsymbol{y}) \equiv \mathcal{N}(\boldsymbol{\theta}_s, \frac{1}{\sigma_y^2} A^{-1} \boldsymbol{X} \boldsymbol{y}, A^{-1}) \tag{7}$$

where $A = \frac{\boldsymbol{X}\boldsymbol{X}^\top}{\sigma_Y^2} + \frac{I_p}{\sigma_\theta^2}$ and the prior $P(\boldsymbol{\theta}_s \mid \boldsymbol{g}_s) \equiv \mathcal{N}(\boldsymbol{\theta}_s; \boldsymbol{0}, \frac{1}{\sigma_\theta^2} I_p)$ (Rasmussen, 2003). The distribution for $\boldsymbol{\theta}_s$ is determined using Bayesian Linear Regression

**Theorem 4.1** (Posterior update rule for the linear SCM)**.** The posterior probability of $\boldsymbol{X}_s$ being the set of parents of $Y$ for a sample $\{\boldsymbol{x}, y\}$ is

$$\log P(\boldsymbol{g}_s \mid \boldsymbol{x}, y) \propto \log P(\boldsymbol{g}_s) - \frac{1}{2}\log(2\pi\sigma_Y^2) - \frac{1}{2}|\Sigma_{\mathrm{prior}}| - \frac{y^2}{2\sigma_Y^2} - \frac{1}{2}\boldsymbol{\mu}_{\mathrm{prior}}\Sigma_{\mathrm{prior}}^{-1}\boldsymbol{\mu}_{\mathrm{prior}}$$

$$+ \frac{1}{2}\boldsymbol{b}^\top C^{-1}\boldsymbol{b} - \frac{1}{2}\log|C| \tag{8}$$

where $\boldsymbol{\theta}_s \mid \boldsymbol{g}_s \sim \mathcal{N}(\boldsymbol{\mu}_{\mathrm{prior}}, \Sigma_{\mathrm{prior}})$, $C = \left(\Sigma_{\mathrm{prior}}^{-1} + \frac{\boldsymbol{x}_s \boldsymbol{x}_s^\top}{\sigma_Y^2}\right)^{-1}$ and $\boldsymbol{b} = \left(\frac{y\boldsymbol{x}_s^\top}{\sigma_Y^2} + \boldsymbol{\mu}_{\mathrm{prior}}^\top \Sigma_{\mathrm{prior}}^{-1}\right) C$

**Proof sketch:** The main step in the proof is

$$P(\boldsymbol{g}_s \mid \boldsymbol{x}, y) = \int_{\boldsymbol{\theta}_s} P(\boldsymbol{g}_s, \boldsymbol{\theta}_s \mid \boldsymbol{x}, y)\mathrm{d}\boldsymbol{\theta}_s = \int_{\boldsymbol{\theta}_s} \frac{P_{\varepsilon_y}(y - \boldsymbol{\theta}_s^\top \boldsymbol{x}_s)P(\boldsymbol{\theta}_s \mid \boldsymbol{g}_s)P(\boldsymbol{g}_s)}{P(\boldsymbol{x}, y)}\mathrm{d}\boldsymbol{\theta}_s \tag{9}$$

In order to solve the integral, we can first drop the denominator. This follows since there are a discrete number of graphs. We can normalize it at the end to get the probabilities. Since both these quantities that depend on $\boldsymbol{\theta}$ are Gaussian, we can solve the integral in closed form. We update these probabilities iteratively, where we use $P(\boldsymbol{g}_s) = P(\boldsymbol{g}_s \mid \boldsymbol{x}, y)$ in the next iteration of the algorithm. The full proof can be found in Appendix B.

We also need to update the parameters for the Bayesian Linear Regression model. Suppose that $\boldsymbol{\theta}_s \sim \mathcal{N}(\boldsymbol{\mu}_{\text{prior}}, \Sigma_{\text{prior}})$, then the update rules are available in closed form. They are $\Sigma_{\text{post}} = \left( \Sigma_{\text{prior}} + \frac{1}{\sigma_Y^2} \boldsymbol{x}_s \boldsymbol{x}_s^\top \right)^{-1}$ and $\boldsymbol{\mu}_{\text{post}} = \Sigma_{\text{post}} \left( \Sigma_{\text{prior}}^{-1} \boldsymbol{\mu}_{\text{prior}} + \frac{y}{\sigma_Y^2} \boldsymbol{x}_s \right)$ Thus, $\boldsymbol{\theta}_s \mid \boldsymbol{g}_s, \boldsymbol{x}, y \sim \mathcal{N}(\boldsymbol{\mu}_{\text{post}}, \Sigma_{\text{post}})$ and this distribution is then the prior in the next iteration of the algorithm.

**Nonlinear Case.** We derived the posterior update rules for the linear case. The problem with this is that the linear SCM is a very rigid assumption, and most problems in practice are non-linear. For the nonlinear case we will use GPs. Suppose we model the causal mechanism of the target variable as

$$f(\boldsymbol{x}_s) \sim \mathcal{GP}(\mathbf{0}, K_{\text{RBF}}(\boldsymbol{x}_s, \boldsymbol{x}_s')). \tag{10}$$

To solve this problem in the nonlinear case, we will use ideas from GPs to project the feature vector $\boldsymbol{x} \in \mathbb{R}^d$ into a higher dimensional space $\phi(\boldsymbol{x}) \in \mathbb{R}^D$, where $D \gg d$ and $D$ can possibly be infinite-dimensional. The rest of the setting stays very similar to the linear case. The RBF kernel is a popular choice for a kernel function because it can approximate any smooth function (Micchelli et al., 2006). However, using this kernel results in infinite-dimensional feature vectors. This means that $Y := \boldsymbol{\theta}_s^\top \phi(\boldsymbol{X}_s) + \varepsilon_Y$ is not available in closed form. GPs do not ever need to compute the vector in the feature space as it can simply leverage the kernel trick (Rasmussen, 2003), which says that $K(\boldsymbol{x}, \boldsymbol{x}') = \phi(\boldsymbol{x})^\top \phi(\boldsymbol{x}')$. However, in our case we do need to compute the higher dimensional vectors as the closed-form expression in Theorem 4.1 contains an outer-product. We will use Fourier transforms to approximate the feature vector of the radial basis kernel function in a lower dimensional space (Rahimi & Recht, 2007). The result we are using is

$$z(\boldsymbol{x}) = \sqrt{\frac{2}{D}} [\cos(\omega_1^\top \boldsymbol{x} + b_1), \ldots, \cos(\omega_D^\top \boldsymbol{x} + b_D)] \tag{11}$$

where $b_i \sim \mathcal{U}(0, 2\pi)$ and $\omega_i \sim \mathcal{N}(\mathbf{0}, \frac{1}{\sigma^2} I_D)$. We can now use this feature vector to approximate the GP and we can derive the posterior distribution is exactly the same way as in Theorem 4.1 with $\boldsymbol{x}_s$ being replaced with $z(\boldsymbol{x}_s)$. The approximation error is bounded (Rahimi & Recht, 2007), which means we can approximate the true underlying function with some error bound. This error bound decays exponentially as the number of dimensions $D$ increase. The parameters are still estimated using Bayesian Linear Regression, but now in this higher dimensional space as discussed in Appendix B.2.

**Prior probabilities over larger graphs.** Currently, the posterior becomes intractable as the number of nodes in the graph increases. This is because we need to compute the posterior for each possible subset of nodes. To address this, we leverage $\mathcal{D}_{\text{obs}}$ to derive an initial prior probability for a set of nodes being the direct parents of $Y$. Specifically, we use the doubly robust causal feature selection methodology, which is effective for identifying direct causal parents of a target variable even in high-dimensional and non-linear settings (Soleymani et al., 2020; Angelis et al., 2023; Quinzan et al., 2023). This approach provides a point estimate of the set of parents. To quantify uncertainty, we utilize bootstrap sampling, running the feature selection method multiple times across different bootstrap samples of $\mathcal{D}_{\text{obs}}$. This approach helps to capture the variability in parent selection, which gives a measure of uncertainty for the different subsets. We use the following test statistic $\chi_j := \mathbb{E}_{(x_j, \boldsymbol{x}_j^c) \sim X_j, \boldsymbol{X}_j^c} \left[ (\mathbb{E}[Y \mid x_j, \boldsymbol{x}_j^c] - \mathbb{E}[Y \mid \boldsymbol{x}_j^c])^2 \right]$ to test whether $X_j$ is a parent of $Y$. In this equation $\boldsymbol{X}_j^c$ is the variables in $\boldsymbol{X}$ that do not contain $X_j$ Angelis et al. (2023) show that $\chi_j \neq 0$ if and only if $X_j$ is a direct parent of $Y$. This tests whether the Average Controlled Direct Effect for a variable $X_j$ is zero or not. The doubly robust-method has a $\sqrt{n}$-consistency guarantee under the listed assumptions (Quinzan et al., 2023), which means the prior estimate will improve if the size of the observational dataset increases. This is a useful property as the observational data is easier to obtain than interventional data. It also scales linearly as the dimensions increase. Furthermore in cases where we have prior beliefs about the direct parents of $Y$ we can naturally incorporate it into this framework, without needing the doubly robust part.

---

**Algorithm 1** Pseudocode for CBO-U

---

1: Start with initial dataset $\mathcal{D}_{\text{obs}}$ and $\mathcal{D}_{\text{int}}$
2: Determine prior probabilities for $P(\boldsymbol{g}_s)$ using $\mathcal{D}_{\text{obs}}$ and fit $P(\boldsymbol{\theta}_s \mid \mathcal{D}_{\text{obs}}) \sim \mathcal{N}(\boldsymbol{\theta}_s; \boldsymbol{\mu}_{\text{prior}}, \Sigma_{\text{prior}})$
3: Update $P(\boldsymbol{g}_s)$ and $P(\boldsymbol{\theta}_s)$ using $\mathcal{D}_{\text{int}}$
4: Set $m(\cdot)$ as the prior mean funciton and $K(\cdot, \cdot)$ as the prior covariance function for each $e \in \mathbf{ES}$
5: **for** iterations $t = 1, 2, \ldots, T$ **do**
6:     Compute $\alpha_e \ \forall e \in \mathbf{ES}$ and obtain optimal set value $\boldsymbol{\xi}_t^*$
7:     Intervene and obtain $\mathcal{D}_{\text{int}}^t$
8:     Update $P(\boldsymbol{g}_s)$ and $P(\boldsymbol{\theta}_s)$ using $\mathcal{D}_{\text{int}}^t$
9:     Let $\mathcal{D}_{\text{int}} = \mathcal{D}_{\text{int}} \cup \mathcal{D}_{\text{int}}^t$
10:    Update $m(\cdot)$ and $K(\cdot, \cdot)$ using $P(\boldsymbol{g})$ and refit GP using $\mathcal{D}_{\text{int}}$
11: **end for**
12: **Return best input in data set**: $\boldsymbol{\xi}^* = \arg\min_{\boldsymbol{\xi}_t} y_t$

---

## 4.2 Causal Bayesian Optimization with Unknown Graphs: Algorithm

We will use the standard GP for CBO as the surrogate model (Aglietti et al., 2020) for CBO-U. Since the graph is unknown, estimating the prior mean and kernel function becomes more difficult. As in Branchini et al. (2023), we will introduce a latent variable $\boldsymbol{g}$, which denotes the unknown set of parent variables. We can then use the law of total expectation and the previously derived posterior distribution to compute the mean and variance as

$$\mathbb{E}[Y \mid \boldsymbol{\xi}, \boldsymbol{C}] = \mathbb{E}_{P(\boldsymbol{g})}\left[\mathbb{E}[Y \mid \boldsymbol{\xi}, \boldsymbol{C}, \boldsymbol{g}]\right] \tag{12}$$

$$\mathbb{V}[Y \mid \boldsymbol{\xi}, \boldsymbol{C}] = \mathbb{V}_{P(\boldsymbol{g})}\left[\mathbb{E}[Y \mid \boldsymbol{\xi}, \boldsymbol{C}, \boldsymbol{g}]\right] + \mathbb{E}_{P(\boldsymbol{g})}\left[\mathbb{V}[Y \mid \boldsymbol{\xi}, \boldsymbol{C}, \boldsymbol{g}]\right] \tag{13}$$

We use $\mathcal{D}_{\text{obs}}$ and do-calculus to estimate the inner expectation, and then the posterior distribution to estimate the final mean and variance function. This will capture the uncertainty over both the graph and the surrogate model. The final algorithm is then given in Algorithm 1. We use the invariance property (Peters et al., 2016) to update the posterior probability with the new interventional data point. It is shown that using interventional data improves the the accuracy of causal effect estimation (Hauser & Bühlmann, 2015).

## 5 Datasets and Experimental Setup

**Datasets.** *Erdos-Renyi.* We evaluate the methodology on randomly generated Erdos-Renyi graphs (Erdo & Renyi, 1959), considering both linear and nonlinear examples. We select a target variable $Y$ at random, and assume that other nodes are manipulative variables. In the first set of experiments, we test the method using a randomly generated Erdos-Renyi graph of size 10, 15, 20 and 50.

*Benchmark experiments.* We utilize the same three examples as in Branchini et al. (2023). Each example is designed to demonstrate both the strengths and limitations of the current methods. The three examples include the following: the *Toy Example*, which uses the same DAG as in Aglietti et al. (2020); the *Healthcare Example* (Ferro et al., 2015; Thompson, 2019), where the objective is to determine the appropriate dosage of statin or aspirin to minimize PSA levels in patients ($Y$); and the *Epidemiology Example* (Branchini et al., 2023; Havercroft & Didelez, 2012), where the goal is to administer doses of two treatments, $T$ and $R$, to reduce the HIV viral load ($Y$).

*Dream experiments.* We apply the CBO-U methodology to another downstream task, evaluating the framework in a semi-synthetic environment using the DREAM gene interaction network as a basis (Greenfield et al., 2010). Specifically, we examine the *E. coli* gene interaction network, which comprises of 10 nodes. These nodes represent genes, and the edges indicate the direct effects of how one gene influences another. The target variable $Y$ is randomly selected, and all other nodes are considered manipulative variables. This task is ideal for CBO as it mimics real-world scenarios where complete genetic information is often unavailable.

**Baselines.** For the *benchmark experiments* we compare the CBO-U approach against CBO with a known causal graph (Aglietti et al., 2020), BO (Jones et al., 1998), and CEO (Branchini et al., 2023). For full implementation details of these algorithms refer to Appendix C.1. For the *Erdos-Renyi* and the *Dream* experiments we test the method against the CBO methodology (Aglietti et al.,

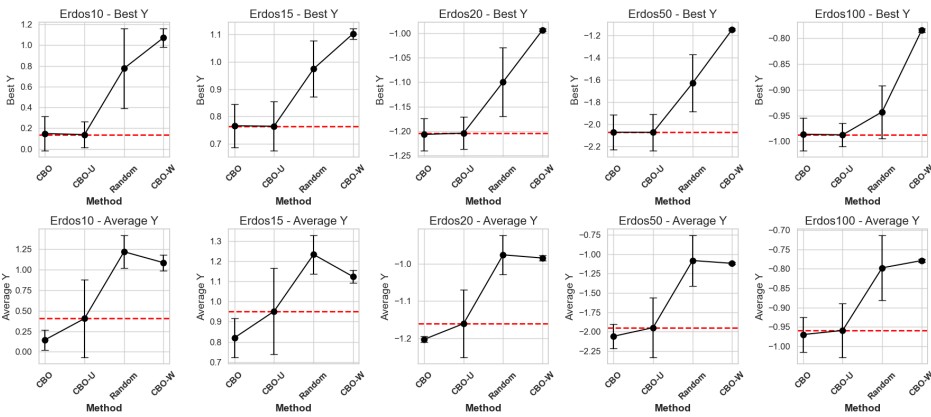

Figure 3: Results on the $Y^* \downarrow$ and the $\bar{Y} \downarrow$ metric across 10 randomly initialized $\mathcal{D}_{\text{obs}}$ and $\mathcal{D}_{\text{int}}$ for randomly generated nonlinear Erdos-Renyi graphs of size 10, 15, 20, 50, 100. Each algorithm was run for 50 trials. The top row shows the results for the $Y^*$ case and the bottom row shows the results for the $\bar{Y}$ case.

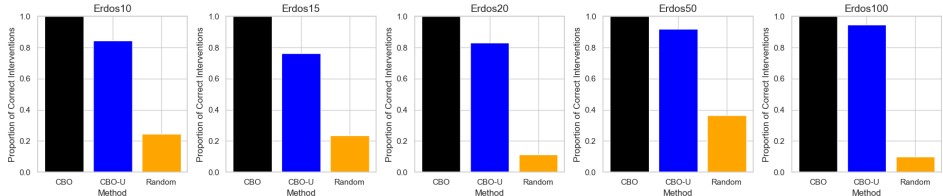

Figure 4: This figure shows the proportion of times each algorithm correctly selected interventions that was a direct parent of the target across 10 different iterations of the algorithm for the nonlinear Erdos-Renyi graphs. CBO-U successfully identifies the optimal interventions the vast majority of the time.

2020) with a correctly specified graph, CBO with wrong graph (CBO-W), and a method that select interventions randomly (Random). Since the CBO methodology has full knowledge of the causal graph, we consider it a performance upper bound. We consider CBO-W and Random as performance lower bounds. We do not consider BO and CEO as neither of the methods can naturally be extended these settings for interventions.

**Evaluation Metrics.** We evaluate the performance of the algorithms by examining the average $Y$ value and the optimal $Y$ selected at each iteration of the algorithm. These two metrics allow for a robust comparison of the different methods. We compare the optimal value that the algorithm converges to, denoted as $Y^*$. If the algorithm converges to a suboptimal target value it will be depicted in this metric. Additionally, we look at the average value selected throughout the run, denoted as $\bar{Y}$. This metric is useful because, as more interventions are performed, the average selected value should decrease (Sussex et al., 2023). This is important in practice because there is a safety aspect to consider; we want to avoid interventions that result in poor outcomes.

**Experimental Setup.** The initial setup is for the CBO algorithm, unless otherwise stated is $N_{\text{obs}} = 200$ and $N_{\text{int}} = 2$. The exploration set is updated after each posterior update, by considering only parent sets with a non-zero posterior distribution of being a direct parent of the target variable $Y$. In the initial experiments we use the causal expected improvement acquistion function (Aglietti et al., 2020).

## 6 RESULTS AND DISCUSSION

**CBO-U scales to larger graphs than existing baselines.** For this experiments we use the Erdos-Renyi setup and the results are given in Figure 3. The CBO-U methodology converges to the same

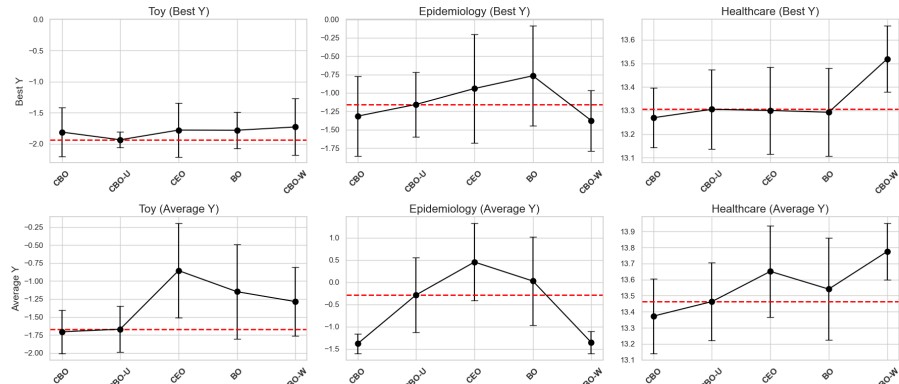

Figure 5: Results on the $Y^*$ ↓ and the $\bar{Y}$ ↓ metric across 10 randomly initialized $\mathcal{D}_{\text{obs}}$ and $\mathcal{D}_{\text{int}}$ for benchmark examples. Each algorithm was run for 30 trials. The top row shows the results for the $Y^*$ case and the bottom row shows the results for the $\bar{Y}$ case. The dashed horizontal line shows the mean of CBO-U which performs better on average than CEO and BO.

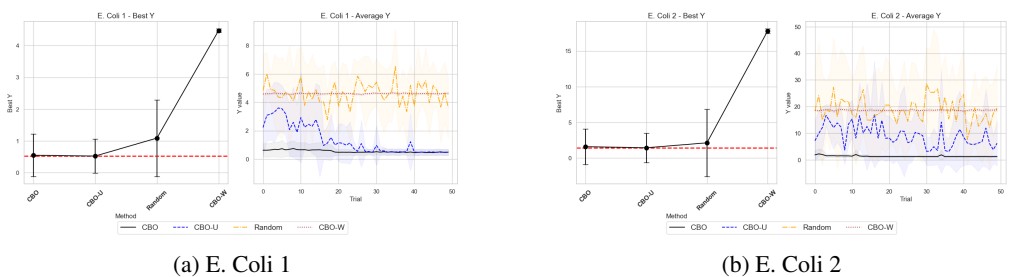

(a) E. Coli 1          (b) E. Coli 2

Figure 6: Comparison of E. Coli 1 and E. Coli 2 in the Dream Interaction network. The figures show both the $Y^*$ ↓ and the $\bar{Y}$ ↓ case at each iteration of the algorithm. Both experiments were across 10 different random initialisation of $\mathcal{D}_{\text{obs}}$ and $\mathcal{D}_{\text{int}}$ and the algorithm was run for 50 trials. CBO-U performs well on real graphs.

optimal value as CBO in all cases, across different graph sizes (10, 15, 20, 50), indicating its effectiveness when the true causal graph is fully unknown. It consistently finds the optimal intervention and value. This underscores its effectiveness. Furthermore in Figure 4 we see that CBO-U consistently intervenes on the different parent variables. These are significant results, as the CBO-U methodology can scale to graphs up to 50 nodes where no prior knowledge of the structure is available. All other methods break down in this scenario. The original CBO requires knowledge of the causal graph, BO struggles in higher-dimensional problems (Wang et al., 2016) and requires interventions on all nodes at each iteration. This is more costly. Furthmore, CEO requires a strong initial hypothesis of the possible DAGs otherwise it is intractable.

**CBO-U converges to the optimal target value in all cases.** This results for these *benchmark experiments* are given in Figure 5. The CBO-U methodology is competitive with CBO when looking at $Y^*$ and outperforms both CEO and BO in the $\bar{Y}$ case. In all three examples, the CBO-U methodology finds a similar optimal point to CBO, meaning it does successfully identify the optimal intervention. While the average case is worse than CBO due to the uncertainty captured by the posterior distribution, it still is competitive with CEO and BO in the average case. This is an important result, as it demonstrates that CBO-U can handle more complex nonlinear scenarios while recovering performance similar to CBO. Furthermore in Figure 3, we also see that CBO-U outperforms Random and CBO-W on average in all cases while converging to the same value as CBO.

**CBO-U translates to real graphs.** We examine the *E. coli* gene interaction network (Figure 6), consisting of 10 variables representing genes, with edges indicating direct gene interactions. These regulatory interactions provide a controlled environment to test the CBO-U approach. The goal is to optimize genetic interventions, simulating real-world scenarios where full genetic information is

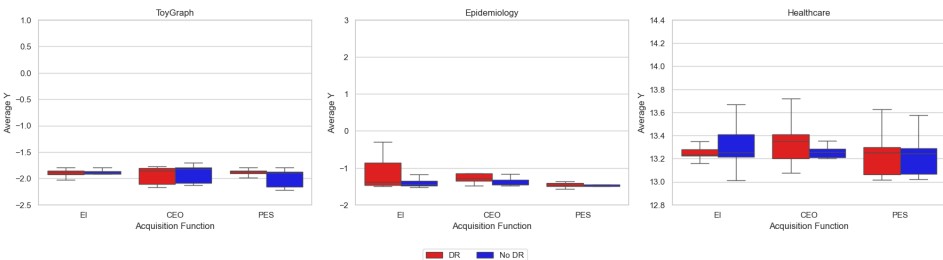

Figure 7: Ablation study with different acquisition functions, and with and without the doubly robust methodology, using 10 random initializations of $\mathcal{D}_{\text{obs}}$ and $\mathcal{D}_{\text{int}}$. The results for $Y^* \downarrow$ are shown. The takeaway is that CBO-U is flexible and performs well across different choices of acquisition functions.

often unavailable. In both cases, CBO-U outperforms Random and CBO-W and converges to the same rate as CBO, which has knowledge of the true causal graph. As the number of iterations increases and more interventional samples are observed, both graph uncertainty and surrogate model uncertainty decrease. This reduction in uncertainty leads to performance that closely matches the case where the causal graph is known. We attribute this to the Bayesian updates, where the interventional data provides more signal for the corresponding edges and improves the posterior of the direct parents.

### 6.1 ABLATIONS

We perform an ablation study (Figure 7) using the simulated Toy, Healthcare and Epidemiology examples to test how the method performs using different acquisition functions and to test how sensitive the method is to the prior specification.

**CBO-U is flexible and can be integrated with different acquisition functions.** We are comparing the causal EI acquisition function (Aglietti et al., 2020), the CEO acquisition function from (Branchini et al., 2023), and the PES acquisition function (Wang & Jegelka, 2017). This further speaks to the generality of the framework, and we can choose an acquisition function to best suit the problem. Different acquisition functions can easily be incorporated into the methodology.

**CBO-U recovers performance with weak prior estimates of the edges**. This means the methodology performs well with both informative and uninformative priors over the direct edges. The doubly robust methodology, however, allows the methodology to scale to larger graphs with more informative priors. We will not be able to tractably update the posterior in cases where we consider all possible subsets in larger graphs.

### 7 CONCLUSION AND FUTURE WORK

**Conclusion.** In this work, we proposed a CBO methodology for cases where the causal graph is unknown that scales to larger graphs. We introduced a novel approach that learns a Bayesian posterior over the direct parents of the target variable. Our theoretical analysis and experiments demonstrated that focusing on direct parents suffices for the CGO problem under some assumptions. We derived a closed-form posterior of the direct parents for the linear case and provided an approximation for the nonlinear case.

**Limitations and Future Work.** The main limitation of our work is the restriction to hard interventions. Future research should focus on extending CBO-U to cases where other interventions can be performed, or to cases where the assumptions of this paper do not hold. The framework is general, and similar approaches should be considered in these cases. For example, Lee & Bareinboim (2018) give optimality conditions in the presence of unobserved confounders and Elahi et al. (2024) show that not all confounders need to be discovered to uncover the possibly optimal intervention set. This is a clear future direction for relaxing Assumption 4.3. One can use such an approach to identify the most relevant part of the causal graph for and then optimize the CBO objective.

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

## A  REPRODUCABILITY STATEMENT

All experiments in this paper were conducted using fixed random seeds to ensure reproducibility of the results. The random seed values used for each experiment are explicitly defined in the provided code. Detailed instructions for running the experiments and verifying the results is included in the anynomous repository: anynomous.4open.science.

## B  THEORETICAL MOTIVATION

### B.1  PROOF OF THEOREM 4.1:

There are three important steps to the proof. They are

**Step 1 – Integrate over $\boldsymbol{\theta}_s$ :**

$$P(\boldsymbol{g}_s \mid \boldsymbol{x}, y) = \int_{\boldsymbol{\theta}_s} P(\boldsymbol{g}_s, \boldsymbol{\theta}_s \mid \boldsymbol{x}, y)\mathrm{d}\boldsymbol{\theta}_s \tag{14}$$

$$= \int_{\boldsymbol{\theta}_s} \frac{P_{\varepsilon_y}(y - \boldsymbol{\theta}_s^\top \boldsymbol{x}_s)P(\boldsymbol{\theta}_s \mid \boldsymbol{g}_s)P(\boldsymbol{g}_s)}{P(\boldsymbol{x}, y)}\mathrm{d}\boldsymbol{\theta}_s \tag{15}$$

**Step 2 – Drop the denominator:**

Since there are a discrete number of graphs, we do not need to consider the denominator. We can normalize it at the end to get the probabilities. We are interested in

$$P(\boldsymbol{g}_s \mid \boldsymbol{x}, y) \propto \int_{\boldsymbol{\theta}_s} P_{\varepsilon_y}(y - \boldsymbol{\theta}_s^\top \boldsymbol{x}_s)P(\boldsymbol{\theta}_s \mid \boldsymbol{g}_s)P(\boldsymbol{g}_s)\mathrm{d}\boldsymbol{\theta}_s \tag{16}$$

$$\propto P(\boldsymbol{g}_s) \int_{\boldsymbol{\theta}_s} \mathcal{N}(y - \boldsymbol{\theta}_s^\top \boldsymbol{x}_s; 0, \sigma_Y^2)\mathcal{N}(\boldsymbol{\theta}; \boldsymbol{\mu}_{\text{prior}}, \Sigma_{\text{prior}})\mathrm{d}\boldsymbol{\theta}_s \tag{17}$$

**Step 3 – The integral of Gaussian distributions:**

Since, both these equations are Gaussian, we can solve the integral in closed form. We use the following property

$$\boldsymbol{X} \sim \mathcal{N}_p(\boldsymbol{\mu}, \Sigma) \text{ then } \int_{\boldsymbol{x}} \mathcal{N}(\boldsymbol{x}; \boldsymbol{\mu}, \Sigma)\mathrm{d}\boldsymbol{x} = 1 \tag{18}$$

We solve the integral by writing $\boldsymbol{\theta}_s$ in the form of a $\mathcal{N}(\boldsymbol{b}, C)$ distribution, where the values for $\boldsymbol{b}$ and $C$ are determined by completing the square in the multivariate case. We consider the part within the integral. If we write out both pdfs

$$P_{\varepsilon_y}(y - \boldsymbol{\theta}_s^\top \boldsymbol{x}_s) = \frac{1}{\sqrt{2\pi\sigma_y^2}} \exp\left(-\frac{1}{2\sigma_y^2}(y - \boldsymbol{\theta}_s^\top \boldsymbol{x}_s)^2\right)$$

$$\propto \exp\left(-\frac{1}{2\sigma_y^2}(y^2 - 2y\boldsymbol{x}_s^\top \boldsymbol{\theta}_s + \boldsymbol{\theta}_s^\top \boldsymbol{x}_s \boldsymbol{x}_s^\top \boldsymbol{\theta}_s)\right)$$

$$P(\boldsymbol{\theta}_s \mid \boldsymbol{g}_s) = (2\pi)^{-p/2}|\Sigma_{\text{prior}}|^{-1/2} \exp\left(-\frac{1}{2}(\boldsymbol{\theta}_s - \boldsymbol{\mu}_{\text{prior}})^\top \Sigma_{\text{prior}}^{-1}(\boldsymbol{\theta}_s - \boldsymbol{\mu}_{\text{prior}})\right)$$

$$\propto \exp\left(-\frac{1}{2}(\boldsymbol{\theta}_s - \boldsymbol{\mu}_{\text{prior}})^\top \Sigma_{\text{prior}}^{-1}(\boldsymbol{\theta}_s - \boldsymbol{\mu}_{\text{prior}})\right)$$

To solve the integral we only, need to consider the terms that depend on $\boldsymbol{\theta}$. The rest we will treat as a constant and add it back at the end

$$P_{\varepsilon_y}(y - \boldsymbol{\theta}_s^\top \boldsymbol{x}_s) \propto \exp\left(\frac{y\boldsymbol{x}_s^\top \boldsymbol{\theta}_s}{\sigma_y^2} - \frac{1}{2\sigma_y^2}\boldsymbol{\theta}_s^\top \boldsymbol{x}_s \boldsymbol{x}_s^\top \boldsymbol{\theta}_s\right)$$

$$P(\boldsymbol{\theta}_s \mid \boldsymbol{g}_s) \propto \exp\left(-\frac{1}{2}\boldsymbol{\theta}_s^\top \Sigma_{\text{prior}}^{-1}\boldsymbol{\theta}_s + \boldsymbol{\mu}_{\text{prior}}^\top \Sigma_{\text{prior}}^{-1}\boldsymbol{\theta}_s\right)$$

Multiplying these two together, the term in the exponent can be written as

$$-\frac{1}{2}\boldsymbol{\theta}_s^\top \left(\Sigma_{\text{prior}}^{-1} + \frac{\boldsymbol{x}_s \boldsymbol{x}_s^\top}{\sigma_y^2}\right) \boldsymbol{\theta}_s + \left(\frac{y\boldsymbol{x}_s^\top}{\sigma_y^2} + \boldsymbol{\mu}_{\text{prior}}^\top \Sigma_{\text{prior}}^{-1}\right)\boldsymbol{\theta}_s \tag{19}$$

In order, to solve this integral we need to write the term in the exponent in terms of the pdf of a multivariate normal distribution and use property 18.

Now, suppose $\boldsymbol{X} \sim \mathcal{N}_p(\boldsymbol{b}, C)$, then

$$P(\boldsymbol{x}) = (2\pi)^{-p/2}|C|^{-1/2}\exp\left(-\frac{1}{2}(\boldsymbol{x}-\boldsymbol{b})^\top C^{-1}(\boldsymbol{x}-\boldsymbol{b})\right)$$

$$= (2\pi)^{-p/2}|C|^{-1/2}\exp\left(-\frac{1}{2}\boldsymbol{x}^\top C^{-1}\boldsymbol{x} + \boldsymbol{b}^\top C^{-1}\boldsymbol{x} - \frac{1}{2}\boldsymbol{b}^\top C^{-1}\boldsymbol{b}\right)$$

Substituting these values into our expression

$$C^{-1} = \left(\Sigma_{\text{prior}}^{-1} + \frac{\boldsymbol{x}_s \boldsymbol{x}_s^\top}{\sigma_y^2}\right)$$

$$\Rightarrow C = \left(\Sigma_{\text{prior}}^{-1} + \frac{\boldsymbol{x}_s \boldsymbol{x}_s^\top}{\sigma_y^2}\right)^{-1}$$

$$\boldsymbol{b}^\top C^{-1} = \left(\frac{y\boldsymbol{x}_s^\top}{\sigma_y^2} + \boldsymbol{\mu}_{\text{prior}}^\top \Sigma_{\text{prior}}^{-1}\right)$$

$$\Rightarrow \boldsymbol{b}^\top = \left(\frac{y\boldsymbol{x}_s^\top}{\sigma_y^2} + \boldsymbol{\mu}_{\text{prior}}^\top \Sigma_{\text{prior}}^{-1}\right) C$$

If we return to 19, then we can add and subtract $\frac{1}{2}\boldsymbol{b}^\top C^{-1}\boldsymbol{b}$ to write the exponent in terms of the pdf of a normal distribution. This means that

$$-\frac{1}{2}\boldsymbol{\theta}_s^\top \left(\Sigma_{\text{prior}}^{-1} + \frac{\boldsymbol{x}_s \boldsymbol{x}_s^\top}{\sigma_y^2}\right)\boldsymbol{\theta}_s + \left(\frac{y\boldsymbol{x}_s^\top}{\sigma_y^2} + \boldsymbol{\mu}_{\text{prior}}^\top \Sigma_{\text{prior}}^{-1}\right)\boldsymbol{\theta}_s$$

$$= -\frac{1}{2}\boldsymbol{\theta}_s^\top C^{-1}\boldsymbol{\theta}_s + \boldsymbol{b}^\top C^{-1}\boldsymbol{\theta}_s - \frac{1}{2}\boldsymbol{b}^\top C^{-1}\boldsymbol{b} + \frac{1}{2}\boldsymbol{b}^\top C^{-1}\boldsymbol{b}$$

$$= -\frac{1}{2}(\boldsymbol{\theta}_s - \boldsymbol{b})^\top C^{-1}(\boldsymbol{\theta}_s - \boldsymbol{b}) + \frac{1}{2}\boldsymbol{b}^\top C^{-1}\boldsymbol{b}$$

And it follows that

$$\int_{\boldsymbol{\theta}_s} \exp\left(-\frac{1}{2}(\boldsymbol{\theta}_s - \boldsymbol{b})^\top C^{-1}(\boldsymbol{\theta}_s - \boldsymbol{b})\right)\mathrm{d}\boldsymbol{\theta}_s = (2\pi)^{p/2}|C|^{-1/2}$$

if we treat the term inside the integral as the pdf of a $\mathcal{N}(\boldsymbol{b}, C)$ distribution.

Using this result, the final expression is now

$$P(\boldsymbol{g}_s \mid \boldsymbol{x}, y) = P(\boldsymbol{g}_s)(2\pi\sigma_y^2)^{-1/2}(2\pi)^{-p/2}|\Sigma_{\text{prior}}|^{-1/2}\cdot$$

$$\exp\left(-\frac{y^2}{2\sigma_y^2} - \frac{1}{2}\boldsymbol{\mu}_{\text{prior}}\Sigma_{\text{prior}}^{-1}\boldsymbol{\mu}_{\text{prior}} + \frac{1}{2}\boldsymbol{b}^\top C^{-1}\boldsymbol{b}\right)(2\pi)^{p/2}|C|^{-1/2}$$

And in the logspace the answer is

$$\log P(\boldsymbol{g}_s \mid \boldsymbol{x}, y) = \log P(\boldsymbol{g}_s) - \frac{1}{2}\log(2\pi\sigma_y^2) - \frac{1}{2}|\Sigma_{\text{prior}}|$$

$$-\frac{y^2}{2\sigma_y^2} - \frac{1}{2}\boldsymbol{\mu}_{\text{prior}}\Sigma_{\text{prior}}^{-1}\boldsymbol{\mu}_{\text{prior}} + \frac{1}{2}\boldsymbol{b}^\top C^{-1}\boldsymbol{b} - \frac{1}{2}\log|C|$$

## B.2 EXTENSION TO THE NONLINEAR CASE

The proof in the nonlinear case follows in exactly the same way as in the linear case.

**Integrate over $\boldsymbol{\theta}_s$ :**

$$
\begin{aligned}
P(\boldsymbol{g}_s \mid \boldsymbol{x}, y) &= \int_{\boldsymbol{\theta}_s} P(\boldsymbol{g}_s, \boldsymbol{\theta}_s \mid \boldsymbol{x}, y) \mathrm{d}\boldsymbol{\theta}_s \\
&= \int_{\boldsymbol{\theta}_s} \frac{P_{\varepsilon_y}(y - \boldsymbol{\theta}_s^\top \phi(\boldsymbol{x}_s)) P(\boldsymbol{\theta}_s \mid \boldsymbol{g}_s) P(\boldsymbol{g}_s)}{P(\boldsymbol{x}, y)} \mathrm{d}\boldsymbol{\theta}_s
\end{aligned}
$$

The difference now, is that there are more parameters in $\boldsymbol{\theta}_s$. But the resulting expressions will be the same, since we are still integrating over multivariate normal distributions. The final expression will be the same where $\boldsymbol{x}_s$ will simply be replaced by $\phi(\boldsymbol{x}_s)$. We then use (11) to approximate the feature vector in a lower dimensional space. The dimensions of $\boldsymbol{\mu}$ and $\boldsymbol{\Sigma}$ will now change to the dimensions of the new feature vectors. And following the same steps the result is

**Posterior approximation.** The approximate posterior probability of $\boldsymbol{X}_s$ being the set of parents of $Y$ for a sample $\{\boldsymbol{x}, y\}$ is

$$
\begin{aligned}
\log P(\boldsymbol{g}_s \mid \boldsymbol{x}, y) \propto \log P(\boldsymbol{g}_s) &- \frac{1}{2}\log(2\pi\sigma_Y^2) - \frac{1}{2}|\Sigma_{\text{prior}}| - \frac{y^2}{2\sigma_Y^2} - \frac{1}{2}\boldsymbol{\mu}_{\text{prior}}\Sigma_{\text{prior}}^{-1}\boldsymbol{\mu}_{\text{prior}} \\
&+ \frac{1}{2}\boldsymbol{b}^\top C^{-1}\boldsymbol{b} - \frac{1}{2}\log|C|
\end{aligned}
$$

where

$$
\boldsymbol{\theta}_s \mid \boldsymbol{g}_s \sim \mathcal{N}(\boldsymbol{\mu}_{\text{prior}}, \Sigma_{\text{prior}})
$$

$$
C = \left(\Sigma_{\text{prior}}^{-1} + \frac{z(\boldsymbol{x}_s)z(\boldsymbol{x}_s)^\top}{\sigma_Y^2}\right)^{-1}
$$

$$
\boldsymbol{b} = \left(\frac{yz(\boldsymbol{x}_s)^\top}{\sigma_Y^2} + \boldsymbol{\mu}_{\text{prior}}^\top \Sigma_{\text{prior}}^{-1}\right) C
$$

The initial probabilities are $\boldsymbol{\mu}_{\text{prior}} = \frac{1}{\sigma_Y^2}A^{-1}z(\boldsymbol{X})\boldsymbol{y}$ and $\Sigma_{\text{prior}} = A^{-1}$ where $A = \frac{z(\boldsymbol{X})z(\boldsymbol{X})^\top}{\sigma_Y^2} + \frac{1}{\sigma_\theta^2}I_D$. This rest of the update rules are the same as in the linear case in the new feature space. Rahimi & Recht (2007) show that this approximation is bounded, and that the bound decays exponentially as the number of features $D$ increase. This means that we can approximate the GP with some error bound. We can now calculate the likelihood of observing a sample using the GP that was fitted with parameters $\boldsymbol{\theta}$.

## B.3 SUFFICIENCY OF LEARNING PARENTS

In Section 4 we validated that hard interventions are optimal for certain cases and the optimal intervention set are the parent set. We specifically solved the problem for these cases. Figure 8 gives an intuitive justification as to why learning the other edges are redundant for this problem.

The main proposition from Gultchin et al. (2023) we are using are:

**Proposition B.1** (Optimality of hard interventions). *In a causal graph $\mathcal{G}$, if $\boldsymbol{pa}_Y \subseteq \boldsymbol{X}$ and $\boldsymbol{sp}_Y = \emptyset$, there an there exists an intervention set $\boldsymbol{X}_s \in \boldsymbol{pa}_Y$ that solves the CGO problem.*

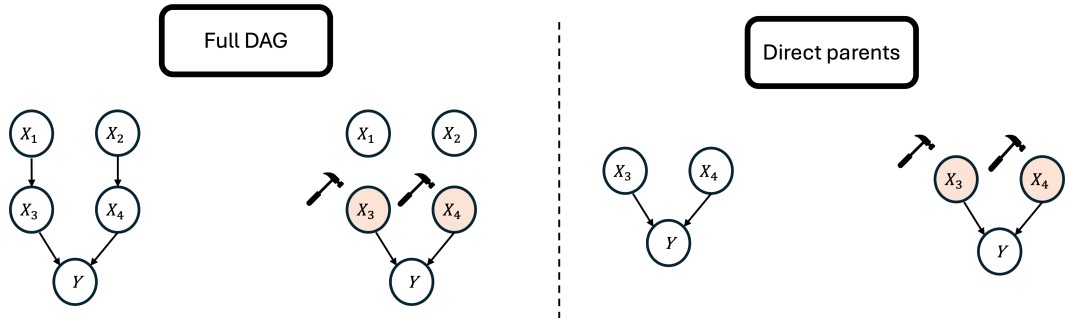

Figure 8: Intuitive justification as to why we are not learning the full graph, due to the equivalence of the interventional distribution.

### B.4 DIRECT PARENTS

In this section, we are going to show how identifying only the direct parents influences the performance of CBO in the healthcare examples. We are going to consider the following cases

1. **Case 1:** Specify full graph
2. **Case 2:** Specify all direct parents
3. **Case 3:** Specify only the direct parents of the manipulative variables.

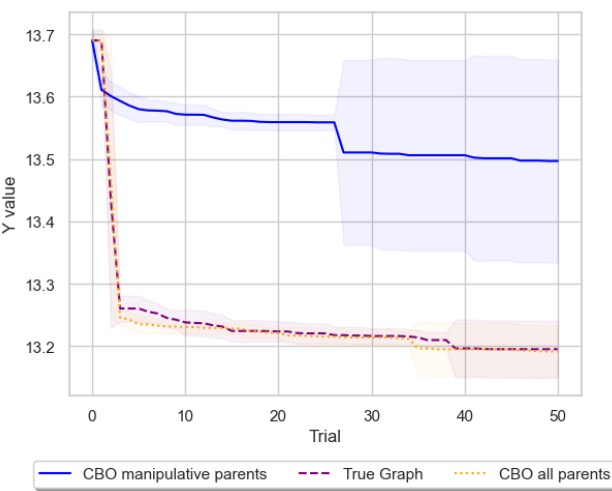

Figure 9: This figures shows the best $Y$ value selected at each iteration of the algorithm for each of the three cases. This was run accross 10 initializations of $\mathcal{D}_{\text{obs}}$ and $\mathcal{D}_{\text{int}}$

Figure 9 shows that we do not need full knowledge for the CBO algorithm to perform well. Additionally it shows that even if we cannot intervene on the direct parents it is still important to specify them correctly due to the assumptions of an SCM.

## C EXPERIMENTAL SETUP

### C.1 BENCHMARK EXPERIMENTS

In these experiments, we compare the BO (Jones et al., 1998), the CBO (Aglietti et al., 2020) and the CEO (Branchini et al., 2023) as baseline algorithms. The training details for each algorithm are

1. In the BO algorithm, we use $f(\boldsymbol{x}) \sim \mathcal{GP}(\mathbf{0}, K_{\text{RBF}})$ as the surrogate model. To optimize the hyperparameters of the GP, we maximize the marginal log likelihood of the observed data,

which allows us to adjust the kernel parameters to best capture the underlying structure of the data and improve the model's predictive performance.

2. In the CBO algorithm, we used the code from Aglietti et al. (2020) as the baseline. We adapted the code, to make it easier to translate to different baselines. The prior $m(\cdot)$ and $K(\cdot, \cdot)$ are estimated using $\mathcal{D}_{\text{obs}}$. For each node in the DAG, we use GP Regression to model the relationship between the node and its parents. The input space for each GP will depend on the number of parents the node has. These GP parameters are optimized by maximizing the marginal log-likelihood. The prior mean and kernel function are then estimated using the *do-calculus* rules.

3. We adapted the CEO codebase from Branchini et al. (2023) to work in the same environments as the other two. The GPs are fitted in the same way as CBO. In this case, to optimize the acquisition function, we have to select a number of anchor points. In each case, we optimize the acquisition function over 35 anchor points.

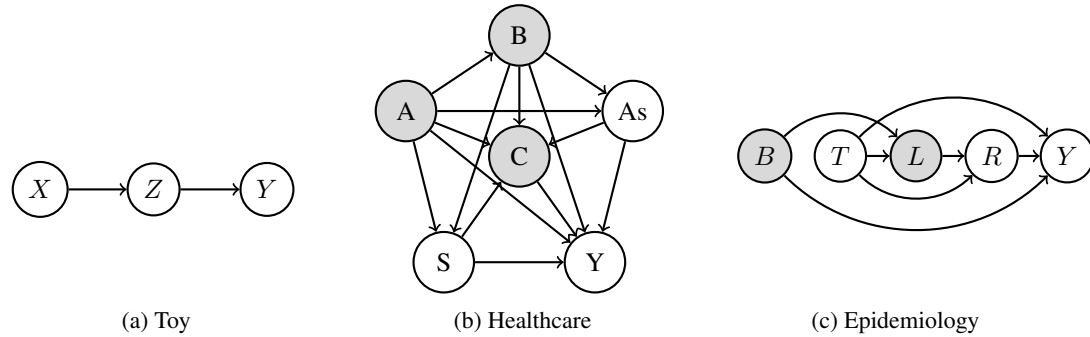

(a) Toy      (b) Healthcare      (c) Epidemiology

Figure 10: Causal DAGs for simulated examples. In each graph, gray nodes represent non-manipulative variables, while white nodes represent manipulative variables.

### C.1.1 TOY

The SEM for this DAG is

$$X = \varepsilon_X \text{ where } \varepsilon_X \sim \mathcal{N}(0, 1)$$
$$Z = 4 + e^{-X} + \varepsilon_Z \text{ where } \varepsilon_Z \sim \mathcal{N}(0, 1)$$
$$Y = \cos(Z) - e^{-Z/20} + \varepsilon_Y \text{ where } \varepsilon_Y \sim \mathcal{N}(0, 1)$$

### C.1.2 HEALTHCARE

The SEM for this DAG is

$$A = \mathcal{U}(55, 75)$$
$$B = 27 - 0.1A + \varepsilon \text{ where } \varepsilon \sim \mathcal{N}(0, 0.7)$$
$$As = \sigma(-0.8 + 0.1A + 0.03B)$$
$$S = \sigma(-13 + 0.1A + 0.2B)$$
$$C = \sigma(2.2 - 0.5A + 0.01B - 0.04S + 0.02As)$$
$$Y = 6.8 + 0.04A - 0.15B - 0.6S + 0.55As + C + \varepsilon \text{ where } \varepsilon \sim \mathcal{N}(0, 0.4)$$

### C.1.3 EPIDEMIOLOGY

The SEM for this DAG is

$$B = \varepsilon_B \text{ where } \varepsilon_B \sim \mathcal{U}(-1, 1)$$
$$T = \varepsilon_T \text{ where } \varepsilon_T \sim \mathcal{U}(4, 8)$$
$$L = \text{expit}(0.5T + B)$$
$$R = 4 + L \cdot T$$
$$Y = 0.5 + \cos(4T) + \sin(-L + 2R) + B + \varepsilon_Y \text{ where } \varepsilon_Y \sim \mathcal{N}(0, 1)$$

### C.2 ERDOS-RENYI EXPERIMENTS

We will evaluate the methodology on randomly generated Erdos-Renyi graphs (Erdo & Renyi, 1959), considering both linear and nonlinear examples. The graph generation process will follow the approach used in Lorch et al. (2021). The process works as follows:

1. For the linear case, the edge weights are uniformly sampled. The edge weights are sampled from a $\mathcal{U}(-5, 5)$ distribution.

2. For the nonlinear case, a neural network will be used to parameterize the mean function of the SCM as a function of its parent nodes. The specific neural network that parameterizes the mean has 5 hidden layers, and the weights are sampled from a $\mathcal{N}(0, 1)$ distribution.

The exploration set is updated after each posterior update, by considering only parent sets with a non-zero posterior distribution of being a direct parent of the target variable $Y$. We will test the method using a randomly generated Erdos-Renyi graph of size 10, 15, 20 and 50. The randomly generated graphs in this example have an expected number of edges of 1 per node. The noise is generated using an isotropic-gaussian distribution, which means the noise is the same for each node.

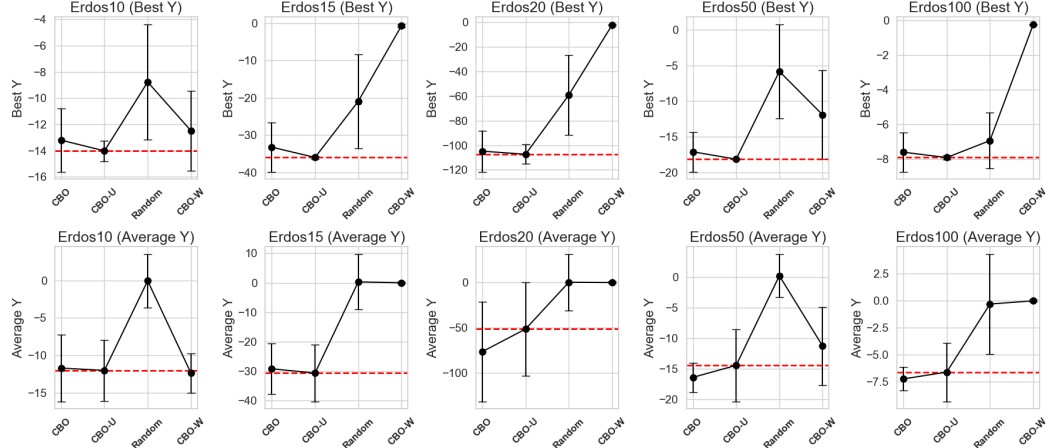

Figure 11: Results on the $Y^* \downarrow$ and the $\bar{Y} \downarrow$ metric across 10 randomly initialized $\mathcal{D}_{\text{obs}}$ and $\mathcal{D}_{\text{int}}$ for randomly generated nonlinear Erdos-Renyi graphs of size 10, 15 and 20. Each algorithm was run for 50 trials. The top row shows the results for the $Y^*$ case and the bottom row shows the results for the $\bar{Y}$ case.

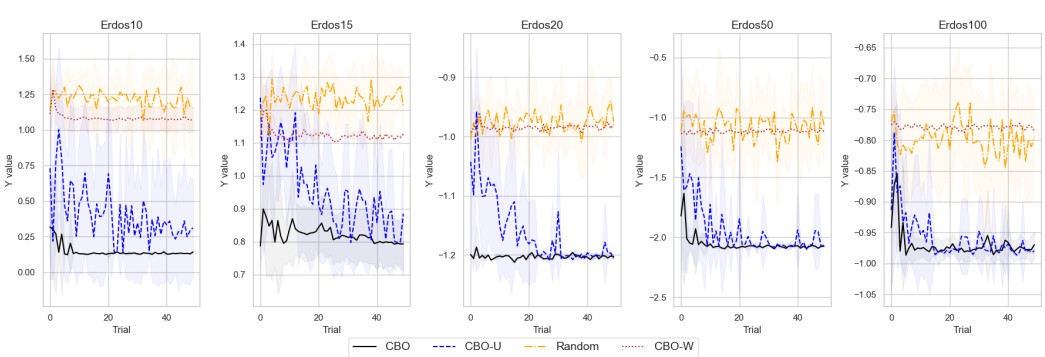

Figure 12: Results on across 10 randomly initialized $\mathcal{D}_{\text{obs}}$ and $\mathcal{D}_{\text{int}}$ for randomly generated nonlinear Erdos-Renyi graphs of size 10, 15 and 20. Each algorithm was run for 50 trials. The figure shows the target value at each iteration of the algorithm and how starts to converge as more are performed.

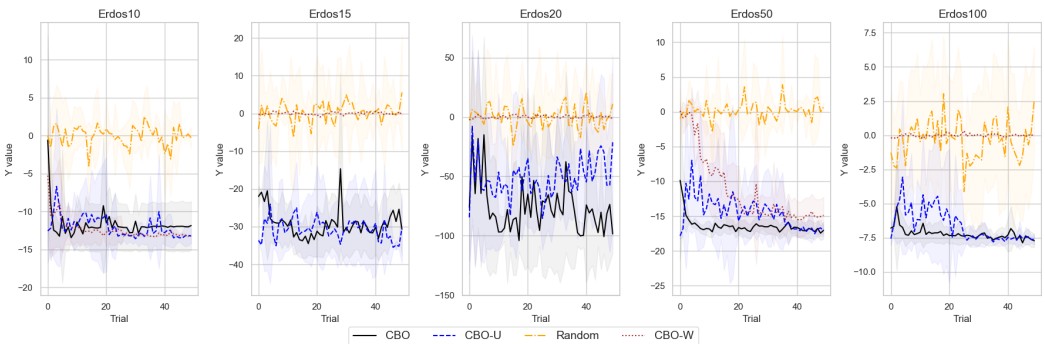

Figure 13: Results on across 10 randomly initialized $\mathcal{D}_{\text{obs}}$ and $\mathcal{D}_{\text{int}}$ for randomly generated nonlinear Erdos-Renyi graphs of size 10, 15, 20 and 50. Each algorithm was run for 50 trials. The figure shows the target value at each iteration of the algorithm and how starts to converge as more are performed.



Figure 14: This figure shows the proportion of times each algorithm correctly selected a interventions that was a direct parent of the target across 10 different iterations of the algorithm for the linear Erdos-Renyi graphs.

## C.3 DREAM EXPERIMENTS

In this section, we provide further evidence of the methodology. In this case, we use the same setup for Dream experiments as in Section 5, but in this case we standardizing the data generating process using the methodology from Ormaniec et al. (2024), and we find similar convergence results.

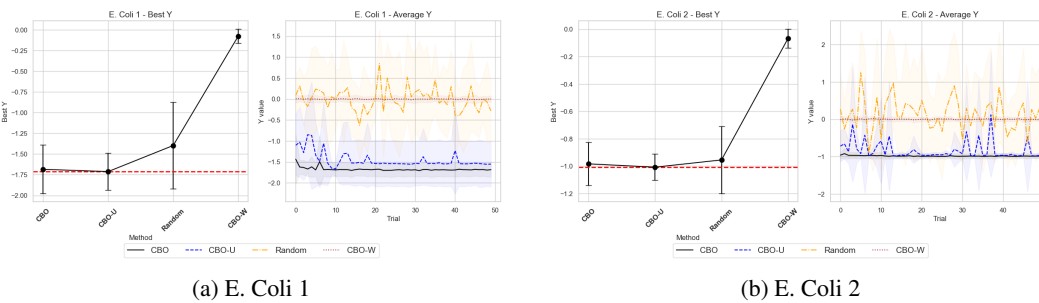

(a) E. Coli 1           (b) E. Coli 2

Figure 15: Comparison of E. Coli 1 and E. Coli 2 in the Dream Interaction network. The figures shows both the $Y^* \downarrow$ and the $\bar{Y} \downarrow$ case at each iteration of the algorithm. Both experiments were accross 10 different random initialisation of $\mathcal{D}_{\text{obs}}$ and $\mathcal{D}_{\text{int}}$ and the algorithm was run across 50 trials.

# D POSTERIOR DISTRIBUTION

## D.1 DOUBLY ROBUST METHODOLOGY

The next design choice pertains to the doubly robust methodology. Within this approach, we must fit two models for $\mathbb{E}[Y \mid \boldsymbol{X}]$ and two models for $\mathbb{E}[Y \mid \boldsymbol{X}_j^c]$, where $\boldsymbol{X}_j^c = \boldsymbol{X} \backslash X_j$. The number of input nodes for these models depends on the number of nodes in the DAG being learned. To maintain consistency across models, we use the same architecture throughout.

1. Specifically, each model is implemented as a multi-layer perceptron (MLP). The neural network is a MLP with three hidden layers, each containing 200 nodes. The ReLU activation function is used between layers.

2. Each model is trained for 500 epochs with a learning rate of 0.01 and a batch size of 32. The Adam optimizer is employed, along with the "Reduce Learning Rate on Plateau" scheduler to adjust the learning rate dynamically.

3. In order to test whether a node is a direct parent of a target variable we use the $\chi_j$ statistic with a T-test and a confidence level of $95\%$. For the purposes of CBO Unknown, we run the doubly methodology with $B = 30$ bootstrap samples of the data. This allows us to estimate the initial sets that are possible parents of the target variable $Y$.

## D.2 POSTERIOR UPDATES

For both the linear and nonlinear ANM, we use $\sigma_Y^2 = 1$ and $\sigma_\theta^2 = 1$ as the prior variances for the observations and parameters, respectively. For the posterior distribution, we scale the data, which justifies the choice of observation variance. We find that the posterior distribution is not sensitive to the specification of $\sigma_\theta^2$. For the nonlinear transformation using the method proposed by Rahimi & Recht (2007) and we set $D = 100$ to approximate the feature space. Additionally we use $\sigma^2 = 1$ and $l = 1$ for the parameters of the radial basis kernel function. We use $\boldsymbol{\theta}$ as the parameters to update.

### D.3 METRICS

### D.4 MEAN ACCURACY

Additionally, we will evaluate the performance of the posterior distribution. Since we are estimating a posterior distribution over the direct parents, we will use a weighted accuracy as the metric for evaluating performance. This weighted accuracy accounts for both the probability of observing a particular set of parents and the accuracy with which the predicted edges match the true edges for each node. It is computed as

$$\text{Mean Accuracy} = \sum_{\boldsymbol{g}} P(\boldsymbol{g})\text{Acc}_{\boldsymbol{g}} \tag{20}$$

$$= \mathbb{E}_{P(\boldsymbol{g})}[\text{Acc}_{\boldsymbol{g}}] \tag{21}$$

where $\text{Acc}_{\boldsymbol{g}}$ is defined as

$$\text{Acc}_{\boldsymbol{g}} = \frac{\sum_{i=1}^{D} I(g_i = g_i^{\text{True}})}{d}. \tag{22}$$

Here, $I(g_i = g_i^{\text{True}})$ is an indicator function that equals 1 if the predicted edge $g_i$ matches the true edge $g_i^{\text{True}}$, and 0 otherwise. The term $d$ represents the number of nodes in the graph.

### D.5 MEAN F1-SCORE

We will also report the mean F1-score. For this metric, we define a true positive (TP) as correctly identifying an edge from a direct parent to the target variable. A false positive (FP) occurs when an edge is predicted where none exists in the true graph. A true negative (TN) is when no edge is predicted, and there is indeed no edge in the true graph. Conversely, a false negative (FN) is when no edge is predicted, but an edge actually exists in the true graph. The F1-score is then calculated as

$$\text{F1} = \frac{2\text{TP}}{2\text{TP} + \text{FP} + \text{FN}} \tag{23}$$

Since, we have a distribution over possible graphs. We compute the mean F1-score as

$$\text{Mean F1} = \sum_{\boldsymbol{g}} P(\boldsymbol{g})\text{F1}_{\boldsymbol{g}} \tag{24}$$

$$= \mathbb{E}_{P(\boldsymbol{g})}[\text{F1}_{\boldsymbol{g}}] \tag{25}$$

where $\text{F1}_{\boldsymbol{g}}$ is the F1-score for $\boldsymbol{g}$.

## D.6 INVESTIGATING THE POSTERIOR PERFORMANCE

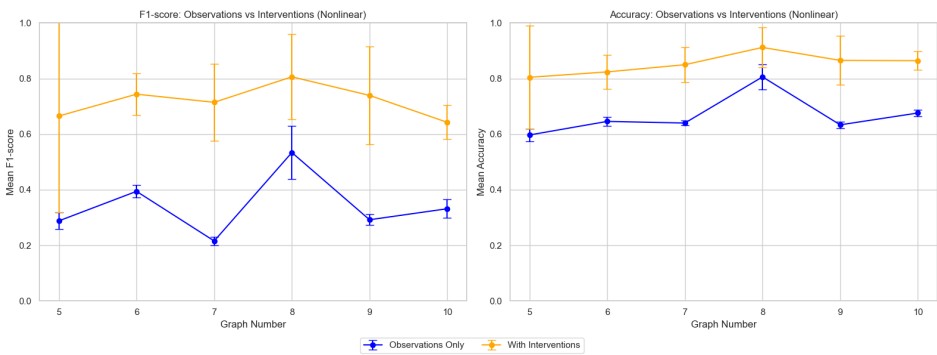

Figure 16: This figure compares, the posterior updates with observational data and the updates with interventional data. The figure shows that the interventional data provides more signal for the posterior updates, which leads to an improved accuracy and f1-score.

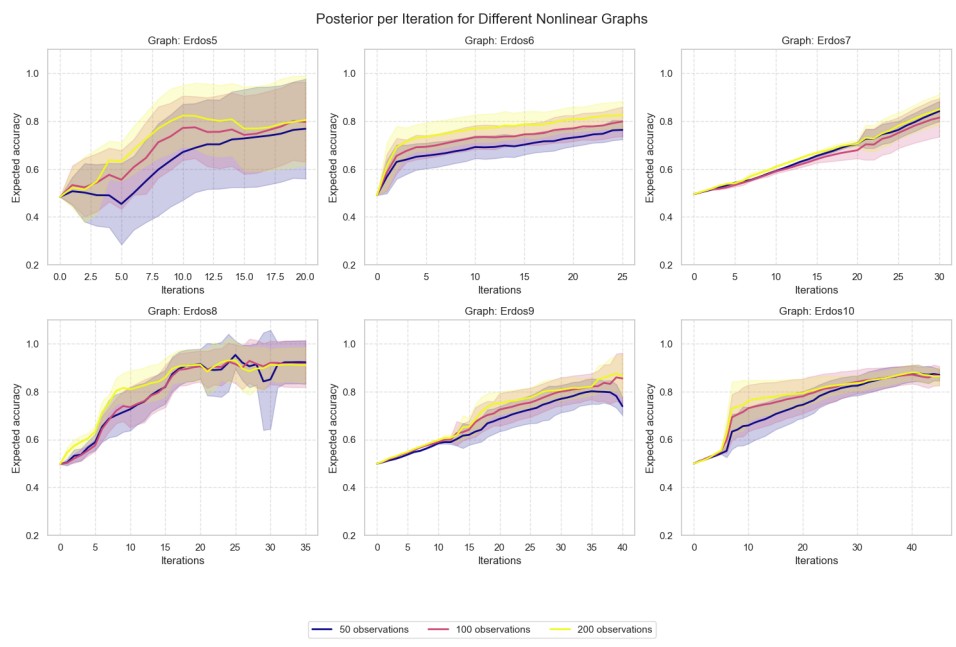

Figure 17: The posterior updates for the nonlinear Erdos-Renyi graph. The figures shows how the mean updates as the number of interventional updates increase. The figure shows the results for the graphs of size $5, 6, 7, 8, 9$ and $10$. The figure shows the results for 10 different random initializations of the observational and the interventional data. The posterior distribution improves at the same rate regardless of the initial size of $\mathcal{D}_{\text{obs}}$

