# OpenReview forum: "Causal Bayesian Optimization with Unknown Causal Graphs"
_ICLR.cc/2025/Conference — Submitted to ICLR 2025_

### Official Review · Reviewer_pmpR · 2024-11-01

**Soundness:** 4
**Presentation:** 4
**Contribution:** 4
**Rating:** 10
**Confidence:** 4

**Summary:**

This paper is a significant contribution to the CBO literate:
- it makes the very practical observation that we do not require the complete graph to do "good" CBO, as previous methods have tackled, but only an idea of the parents of the target variable
- as a consequence, the method can handle bigger larger graphs which is a substantial contribution, see - [ ] 466 which has the strongest statements

**Strengths:**

- authors have a strong grasp of both causal literature, as well as CBO literature which helps exposition of their whole idea
- execution is done well
-- introduction is informative
-- lit review and preliminaries are weighted well
-- methods are clearly introduced and explained
-- experiments are well chosen and support the results well

most importantly, limitations are transparently stated which is of huge importance in the causal inference literature, and science in general

e.g. line 190 appreciate clarity

**Weaknesses:**

I found it hard to find weaknesses in this paper, the idea is clear, the execution is excellent, and the presentation and writing is very effective. It makes a very valid point that previous literature had somewhat overlooked.

Regardless, one strong initial confusion was how this paper is different to the previous literature.

The key sentence is only made in 056 which for the 'fast reader' could easily be missed.

E.g. the abstract does mention the idea, but it's ambiguous: "We demonstrate through theoretical analysis and empirical validation that focusing on the direct causal parents of the target variable is sufficient for optimization." There are other methods that focus on the parental set, and so the contribution of this paper is not immediately clear.

This statement would be possibly better phrased as "We demonstrate through theoretical analysis and empirical validation that focusing ONLY/PRIMARILY/EXCLUSIVELY etc on the direct causal parents of the target variable is sufficient for optimization."

Otherwise, see questions.

**Questions:**

- [ ] Figure 3: what’s red line (apologies if I missed the definition)
- [ ] Fig 6 why are "CBO" and "random" straight lines in the right hand side plots (trials as x-axis)

typos:
- [ ] 215/216 repetition
- [ ] Figure 4 typo ‘an interventions’
- [ ] 421 typo
- [ ] 460 typo

---

> ### Author Response · Authors · 2024-11-18
> **Response to Reviewer 7264**
>
> Thank you for the feedback and for the positive review. We address the questions as follows
>
> - The red line is the performance of the CBO-U method. This was just to make it easier to compare our method with the other methods.
> - In the CBO case, the model has exact knowledge of the causal structure and found the optimal interventions after very few iterations. This happens quite often in CBO, as the surrogate model tends to fit data generating process pretty well. This is a strong motivation for using this methodology. Furthermore, CBO-W also has a straight line since the graph is misspecified, it only selects uninformative interventions, meaning it selects intervention values that does not influence the target value, and the algorithm gets stuck in this minimum. The algorithm will have no way of getting out this local minimum. This further motivates why we incorporated uncertainty into the algorithm to ensure that this does not happen in our proposed solution. (The random baseline is the one that moves around a lot)
>
> We are happy to answer any remaining questions you might have.

---

> > ### Comment · Reviewer_pmpR · 2024-11-20
> > **Thanks for the quick reply**
> >
> > Will keep my rating at 10.

---

### Official Review · Reviewer_qxLe · 2024-11-03

**Soundness:** 3
**Presentation:** 3
**Contribution:** 2
**Rating:** 3
**Confidence:** 4

**Summary:**

The paper presents a novel Causal Bayesian Optimization (CBO) method that operates without a fully known causal graph, focusing on learning only the direct causal parents of a target variable. They propose posterior update rules for the linear case with additive Gaussian noise. For the nonlinear case, they suggest using a Gaussian Process (GP) surrogate model. The authors empirically demonstrate that the proposed method achieves the same optimal target value on synthetic and semi-synthetic causal graphs as methods with fully specified causal graphs.

**Strengths:**

The main contributions of the paper include deriving a closed-form posterior update rule for additive Structural Causal Models (SCMs) with Gaussian noise and formalizing Causal Bayesian Optimization (CBO) for scenarios with unknown causal graphs. For the nonlinear case, the authors propose using Gaussian Processes (GPs). Additionally, they present a scalable solution for handling larger graphs by deriving an initial prior probability for nodes being direct parents of the outcome using observational data.

**Weaknesses:**

The paper has some novelty in the formalism of causal Bayesian optimization with an unknown causal graph. However, I am not entirely convinced of the novelty of the theoretical results in Section 4.1. My primary concern is that there is already existing work on Bayesian causal discovery for linear SCMs with additive Gaussian noise. Since identifying the parents  is a simpler problem, it seems possible that existing approaches could be directly applied for Bayesian optimization purposes. I am referring to the following papers:

https://proceedings.neurips.cc/paper_files/paper/2022/file/675e371eeeea99551ce47797ed6ed33e-Paper-Conference.pdf

https://arxiv.org/pdf/2307.13917

https://proceedings.neurips.cc/paper_files/paper/2021/file/39799c18791e8d7eb29704fc5bc04ac8-Paper.pdf

I would appreciate it if the authors could clarify what makes the results presented in Section 4.1 non-trivial, especially in light of the existing literature. I can consider increasing the score in this case. Additionally, I have concerns regarding some of the assumptions made in the paper, which I have mentioned in the questions section.

**Questions:**

I have few questions for the Authors:

* Can the authors explain why Assumption 4.1 is needed for the proposed solution to work?

* The paper by Lee & Bareinboim (2018), which the authors cite, also provides a complete characterization of possible optimal arms for arbitrary causal graphs with confounders in Proposition 4 and Theorem 6. Can the authors comment on why this result cannot be used to relax Assumption 4.3?

* Additionally, regarding the literature on causal discovery for linear SCMs with additive Gaussian noise, how do the results in the paper compare with them, and what is the novelty of the results in Section 4.1?

---

> ### Author Response · Authors · 2024-11-18
> **Response to Reviewer qxLe**
>
> Thank you for the review and the feedback. We address the questions as follows:
>
> - **Question about novelty:** We believe the novelty of the approach is two-fold. The first novel insights we made is that we do not need full knowledge of the causal graph for CBO. This makes the structure learning aspect of the problem easier, and allows our method to adapt to these general settings. This is where the novelty of Section 4.1 comes in as the results are derived specifically for this problem based on this insight. It incorporates uncertainty and it extends to nonlinear cases. Although there is a range of work with regards to Bayesian Causal Discovery for the full graph, these approaches do not fit into the framework as well as the results presented in this section. For example in those cases, each time an interventional sample is obtained, the full optimization process needs to be run again to determine the new probabilities of all the edges. This is wasteful especially for larger graphs. Our approach only requires simple Bayesian updates and focuses on relevant edges at each iteration of the algorithm. If we were to include methods, we believe a better approach would be to use these methods to determine the prior probabilities over all the edges, and then still use the update rules from Section 4.1 for the algorithm to determine the direct parents. This will also require us to adjust our assumptions based on the assumptions for these methods. This includes causal sufficiency.
> - **Question about Assumption 4.1:** Assumption 4.1 is needed as distinguishing parents and children from observational data alone is not possible and the doubly robust methodology requires this assumption for the theoretical guarantees. This is nicely shown in [1]. However, if we have some prior beliefs about some of the edges and not sure whether it is a child or a parent, then the rest of the theory in Section 4.1 will still hold, and the interventional samples should be able to distinguish between parents and children if we intervene on the children.
> - We addressed this as a general question to all reviewers. Although [2] gives some conditions for optimal intervention in the case of confounders, the bigger challenge lies in the causal discovery aspect of the algorithm. We will now need to learn a slightly larger part of the causal graph in the presence of unobserved confounders which is a very challenging problem in causal discovery.
> - **Question about Assumption 4.3:** We addressed this as a general question to all reviewers. Although [2] gives some conditions for optimal intervention in the case of confounders, the bigger challenge lies in the causal discovery aspect of the algorithm. We will now need to learn a slightly larger part of the causal graph in the presence of unobserved confounders.
> - **Questions about other methods:** To further illustrate why we did not include these types of algorithms we added another experiment. As an extreme case, we included a graph with 100 nodes in the experiments section. Our method is still able to perform optimization. The other methods will try to learn a full causal graph, where most of the edges are not relevant to this optimization problem.
>
> [1] Francesco Quinzan, Ashkan Soleymani, Patrick Jaillet, Cristian R Rojas, and Stefan Bauer. Drcfs: Doubly robust causal feature selection. In International Conference on Machine Learning, pp. 28468–28491. PMLR, 2023.
> [2] Sanghack Lee and Elias Bareinboim. Structural causal bandits: Where to intervene? Advances in neural information processing systems, 31, 2018.
>
> We hope this addresses your concern and we are happy to answer any further questions.

---

> > ### Comment · Reviewer_qxLe · 2024-11-23
> > **Re.**
> >
> > Thank you to the authors for their response. However, I am still not entirely convinced by the novelty of the results in Section 4.1. Under Assumption 4.3, where the reward is not influenced by confounders, the optimal intervention is over the parent set. This is a well-known result from prior work on causal bandits (e.g., https://papers.nips.cc/paper_files/paper/2018/file/c0a271bc0ecb776a094786474322cb82-Paper.pdf).
> >
> > Therefore, the claim that the **insight about not requiring full knowledge of the causal graph for CBO is novel** is not entirely correct. As I pointed out earlier, the problem reduces to learning the parents of the reward node, and there is already existing work on Bayesian causal discovery for linear SCMs with additive Gaussian noise. Since identifying the parent set is a simpler problem, it seems plausible that existing approaches could be directly applied to Bayesian optimization.
> >
> > Overall, I believe the paper does not have  sufficient novelty to warrant an acceptance decision.

---

> > > ### Author Response · Authors · 2024-11-24
> > > **Further response**
> > >
> > > We thank reviewer qxLe for their response and for furthering the discussion. Bayesian Causal Discovery (BCD), while could be used, address a much harder problem which is unnecessary for our setting. Hence, the BCD uncertainty estimates are shown to be much worse [1]. A poor uncertainty estimate leads to poor exploration and low sample efficiency. In light of these insights, and inspired by methods in approximate inference for causal discovery, we only get uncertainty over parent set. Though uncertainty estimates over parents might be an easier problem, it is not explicitly addressed before. It not only allows us to scale but also enables better sample efficiency, as seen in the results. Our central contribution in this paper is about how one could practically and computationally address CBO problem under unknown graphs while getting better uncertainty estimates over parent sets. While simple, we believe that our contribution and empirical results motivates further work in this important and practical setting.
> > >
> > > [1] Karimi Mamaghan, Amir Mohammad, et al. “Challenges and Considerations in the Evaluation of Bayesian Causal Discovery.” In ICML, (2024).

---

> ### Comment · Reviewer_qxLe · 2024-11-25
> **Re. (Edited)**
>
> Thank you to the authors for their response. I acknowledge the contribution of this paper, which addresses the Causal Bayesian Optimization (CBO) problem under unknown graphs using uncertainty estimates over parent sets. The empirical results presented are appreciated.
>
> I have also reviewed the authors’ responses and the comments of other reviewers. The authors noted that the main challenge lies in learning the Markov Blanket of Y in the presence of confounders. However, this observation is not entirely accurate, as when the reward is confounded with any of its ancestors, the candidate optimal intervention targets are specific subsets of the reward's ancestors. The paper titled "Structural Causal Bandits: Where to Intervene?" provides a graphical characterization of the potentially optimal intervention targets in such cases. Including a discussion on this point would enhance the paper.
>
> Upon re-evaluating, I am lowering my score to reject, as I believe the paper lacks sufficient contributions to meet the standards required for acceptance. I also want to emphasize that the numerical score is only a guideline, and my detailed comments should be given greater consideration in the final decision.

---

> > ### Author Response · Authors · 2024-11-29
> > **Response**
> >
> > Hi, thanks for your comment and for continuing the discussion. Yes you are correct, you would need to learn a slightly larger part of the causal graph than the Markov Blanket. Thanks for that observations and we apologize for that incorrect statement in the rebuttal. The main point we were trying to get across is that even in those cases one would not need to learn the full graph, which the CBO framework to work in these cases as well.
> >
> > We just want to make a further point and emphasize that this was not the main purpose of this paper and was only considered as future work. All the theory and results in our paper remains correct.

---

### Official Review · Reviewer_1y9W · 2024-11-03

**Soundness:** 4
**Presentation:** 4
**Contribution:** 3
**Rating:** 8
**Confidence:** 4

**Summary:**

The authors study causal Bayesian optimization (CBO) with an initially completely unknown causal graph. They apply sufficient graphical assumptions for hard interventions (Gultchin et al., 2023) across the parents (Lee and Bareinboim, 2018) of the reward variable $Y$ to be optimal.

To ensure tractability, the support over plausible parents of $Y$ is restricted by the observational dataset using bootstrapped doubly robust causal feature selection. The average causal effect (ACE) of interventions over the plausible parents on $Y$ is modelled by a Gaussian process over intervened datasets.

The authors' approach applies to models where $Y = f ( \mathrm P \mathrm a_Y ) + \epsilon_Y$ is subject to additive, Gaussian noise $\epsilon_Y$. For linear $Y$ with Gaussian noise, a closed form solution for the Bayesian update rule over the linear coefficients $\theta_{\mathrm{Pa}_Y}$ (and thus the ACE) is given. An approximation using the kernel trick is applied for general $f(\cdot)$.

**Strengths:**

The paper is written very well and the authors' contributions, underlying assumptions and limitations are made clear. The performance of the authors' method was evaluated with synthetic datasets and real(?) gene expression datasets. The authors' contributions are high impact and possible theoretical extensions are also clear.

**Weaknesses:**

I do not have many weaknesses to discuss.

Presumably the authors' approach becomes suboptimal when the ground truth parents of $Y$ are not ``discovered'' in the feature selection stage. Was this kind of error rate (a type 2 error rate) measured by the authors at any point? Figure 4 appears to show an error rate of variables falsely included as parents of $Y$ but not those falsely excluded. A breakdown of the underperformance of CBO-U due to type 1 and type 2 errors would give more detailed insight into the limitations of the authors' approach.

Minor comments:

(Section 4) The notation $X_j^c$ for the complement of $X_j$ is used when defining the doubly robust estimator, but the notation is not introduced anywhere in the main text.

(Section 5) The authors appear to be referencing causal expected optimization (CEO) without having defined this in the main text.

(Section 5) In the experiments, were the authors referring to the DREAM5 challenge: i.e., a real-world dataset or earlier synthetic datasets such as DREAM4?

**Questions:**

(Selecting the plausible parents) Have the authors considered an approach to combine the observational and interventional datasets during feature selection? If $X_j$ is not a parent of $Y$ then $\chi_j$ will equal zero across *all* datasets, interventional and observational, due to Assumption 4.3. Would performing the test over pooled data yield better type 1 and 2 error rates?

(Relaxed assumptions) What approaches might one take for a CBO-U extension in cases in which the optimal interventions are soft interventions? Are there similar sufficient conditions under which the optimal interventions are hard interventions but where Assumption 4.3 is violated. The authors' future work section is somewhat terse and it may help to discuss open questions more thoroughly.

---

> ### Author Response · Authors · 2024-11-18
> **Response to Reviewer1y9W**
>
> Thank you for the review. We appreciate the positive feedback and the suggestions. We address the questions as follows:
>
> **Algorithm questions:**
>
> - **Question about type 1 vs type 2 errors**: Yes, you are correct the approach does require the groundtruth parents to be discovered during the initial phase. This is why we used the Bayesian approach with uncertainty estimation. It reduces the chances of not ‘discovering’ the correct parents during the initial phase. It also then updates as it gets new interventional samples. The posterior distribution will then become tighter as the number of samples increases. Currently, the CBO-W shows what will happen if the correct parents are not identified in the algorithm. This essentially shows how the algorithm will get stuck if the correct parents are not determined. This shows what will happen if we cannot determine the direct parents. We will work on incorporating a breakdown when this happens.
> - **Question about the Dream graphs**: We are referring the Dream4 causal graph.
> - **Selecting the plausible parents**: Algorithmically it would not be an issue to use an approach like this, especially if we have both interventional data and observational data available. The reason we did not go that route is that often in practice in this type of setup we would not have interventional samples available or very few interventional samples. Thus, using an approach that works well on observational data made more sense. Furthermore, the goal of CBO-U being to find the optimal intervention in as few interventions as possible. Thus we want to collect the interventional samples in a very targeted manner right from the start and then update the probability based on these samples.
>
> **Assumption questions**:
> - We made the discussion about the limitations clearer in that section. We believe the general framework in those cases will be similar to the one we proposed here. You would want to learn the posterior over the relevant part of the causal graph, and then select interventions in a targeted manner. In those cases, the structure learning aspect will become more challenging.
> - The question about Assumption 4.3 we addressed as a general question to all reviewers. The causal discovery aspect of the algorithm will become more challenging, since we the parents will not necessarily be the optimal intervention set. In this case, we still do not need to learn the full causal graph and we would need to learn the Markov Blanket for the target variable in the presence of confounders. When we relax Assumption 4.3 there are similar conditions we can exploit as referred to in this paper by [1]. This could be a nice follow-up from this paper. The approach can then work in a similar way to CBO-U. Learning this type of structure is however a very challenging problem in causal discovery.
> - There are other approaches for CBO with soft interventions [2, 3]. Both these methods require full knowledge of the graph, but do not have optimality conditions. [3] is a model-based approach and [2] shows cases when these contextual interventions are better than hard interventions. Thus, work needs to be done on determining optimal interventions in this case. One could consider a joint structure-learning and model-based approach.
>
> [1] Sanghack Lee and Elias Bareinboim. Structural causal bandits: Where to intervene? Advances in neural information processing systems, 31, 2018.
>
> [2] Limor Gultchin, Virginia Aglietti, Alexis Bellot, and Silvia Chiappa. Functional causal bayesian
> optimization. In Uncertainty in Artificial Intelligence, pp. 756–765. PMLR, 2023.
>
> [3] Scott Sussex, Anastasiia Makarova, and Andreas Krause. Model-based causal bayesian optimization. International Conference on Learning Representations, 2023
>
> We hope this answers your questions and we are happy to answer any further questions.

---

> > ### Comment · Reviewer_1y9W · 2024-11-19
> >
> > Many thanks to the authors for their reply. I continue to recommend that this paper be accepted.
> >
> > On possible future work for relaxing assumption 4.3., the authors should include Qasim Elahi et al. (2024) as a relevant reference.  Qasim Elahi et al. (2024) show that not all hidden confounders need to discovered to uncover the possibly optimal minimal intervention sets. Similarly to Lee and Bareinboim (2018), their paper exploits a semi-Markov assumption.
> >
> > Muhammad Qasim Elahi, Mahsa Ghasemi and Murat Kocaoglu, Partial Structure Discovery is Sufficient for No-regret Learning in Causal Bandits, 2024, https://arxiv.org/abs/2411.04054.
> >
> > Sanghack Lee and Elias Bareinboim. Structural causal bandits: Where to intervene? NeurIPS, 31, 2018.

---

### Official Review · Reviewer_LGpi · 2024-11-03

**Soundness:** 3
**Presentation:** 3
**Contribution:** 2
**Rating:** 5
**Confidence:** 3

**Summary:**

This paper addresses the problem of causal Bayesian optimization when the causal graph is unknown. Under certain assumptions, the authors propose an algorithm for linear and non-linear structural causal models (SCMs). The approach simultaneously learns the Bayesian posterior over the parent variables (where the best action lies) and optimizes the target variable. The effectiveness of the approach is demonstrated through various experiments.

**Strengths:**

1- The paper addresses an important and challenging problem in causal Bayesian optimization, particularly under the realistic scenario where the causal graph is unknown.

**Weaknesses:**

1- It would be beneficial to review relevant work in the causal bandits literature, where the setting is similar, and some studies address scenarios without a known causal graph.

2- Assumption 4.3 is quite restrictive, limiting the approach’s applicability to graphs without confounders between $Y$ and its ancestors.

**Questions:**

1- I didn’t understand Equation 2—why is it conditioned on $C$? Is not $E[Y | \xi, G]$ the target?


2- As the authors mentioned, previous work has shown that the optimal action is an intervention on parent sets, which is why the focus is on parents in similar settings. My question is, how does your approach perform when there is a confounder between $Y$ and its ancestors? Specifically, how does it handle cases where Assumption 4.3 does not hold?

Typos:

Line 167: $P(Y \vert X_s = \nu, C)$

Line 215: “For the algorithm” is repeated.

Line 251: Missing a period at the end of the sentence.

---

> ### Author Response · Authors · 2024-11-18
> **Response to Reviewer LGpi**
>
> Thank you for the review. We appreciate the feedback and we address the questions as follows.
>
> - **Question 1 about the bandit literature**: Thank you for the suggestion, we have updated our related work with a short section about causal bandit literature.
> -  **Question about Equation 2**: $C$ (which could be $\emptyset$) refers to non-manipulative variables. We want to minimize the expectation with regards to the interventional distribution of the target variable. $C$ is included since $Y$ can be influenced by both the manipulative variables and the non-manipulative variables. This is essentially why the interventional distribution of Y is written as $P(Y | \varepsilon, C)$. Since $Y$ can be influenced by $C$ it is included in the expectation. $G$ is included in the expectation as we do not know the true graph and we need to learn it.
> - **Question about Assumption 4.3**: We addressed this question as a general answer to all reviewers. If we relax this assumption, we can still use results from [1], but in this case a slightly larger part of the graph needs to be learned. This is the main challenge when relaxing this assumption as causal sufficiency is often an underlying assumption in causal discovery literature.
>
> [1] Sanghack Lee and Elias Bareinboim. Structural causal bandits: Where to intervene? Advances in neural information processing systems, 31, 2018.
>
> We hope this addresses your concerns and we would be happy to answer any further questions.

---

> ### Comment · Reviewer_LGpi · 2024-11-25
>
> Thank you for your response. I have read the comments from all reviewers, particularly reviewer qxLe. I agree with reviewer qxLe regarding the contribution of the paper and believe it could be improved by relaxing Assumption 4.3. The findings in recent work [1] may be helpful in this regard. I also agree with reviewer qxLe that the main challenge of the problem in the presence of confounders is not learning Markov Blanket of $Y$.
>
> [1] Muhammad Qasim Elahi, Mahsa Ghasemi and Murat Kocaoglu, Partial Structure Discovery is Sufficient for No-regret Learning in Causal Bandits, 2024, https://arxiv.org/abs/2411.04054.

---

### Author Response · Authors · 2024-11-18
**Overall comment about Assumption 4.3**

Several reviewers raised questions about Assumption 4.3, so we posted a general comment here to emphasize why this assumption was needed and what would happen if it were relaxed.

We believe our approach and framework are quite general, and a similar approach can be used when Assumption 4.3 does not hold. If this assumption does not hold, it is no longer guaranteed that the parent set is the optimal intervention set. We will however still be able to exploit some properties of the causal graph, so we do not need to learn the full graph. We essentially need to know the Markov blanket of Y in order to ensure optimality. The main challenge if this assumption does not hold, is learning the Markov Blanket of Y in the presence of confounders. This is an extremely challenging causal discovery problem, as causal sufficiency is an important assumption in causal discovery literature. To the best of our knowledge, the methods that relax this assumption [1, 2, 3, 4, 5] struggle on small samples or the theoretical results are not general. We do however believe that even with this assumption, our approach is less restrictive than CBO approaches in existing literature. For example, all these CBO methods require full or partial knowledge of the causal graph [6, 7, 8, 9, 10, 11], and our main contribution is relaxing this assumption.

References

[1] David Kaltenpoth and Jilles Vreeken. Causal discovery with hidden confounders using the algorithmic Markov condition. In Proceedings of the Thirty-Ninth Conference on Uncertainty in Artificial Intelligence, volume 216 of Proceedings of Machine Learning Research, pp. 1016–1026. PMLR, 2023.

[2] Juan Miguel Ogarrio, Peter Spirtes, and Joe Ramsey. A hybrid causal search algorithm for latent variable models. In Conference on probabilistic graphical models, pp. 368–379. PMLR,
2016.

[3] Diego Colombo, Marloes H Maathuis, Markus Kalisch, and Thomas S Richardson.
Learning high-dimensional directed acyclic graphs with latent and selection variables. The Annals of Statistics, pp. 294–321, 2012.

[4] Peter L Spirtes, Christopher Meek, and Thomas S Richardson. Causal inference in the presence of latent variables and selection bias. arXiv preprint arXiv:1302.4983, 2013.

[5]  Tom Claassen, Joris Mooij, and Tom Heskes. Learning sparse causal models is not np-hard. arXiv preprint arXiv:1309.6824, 2013.
[6] Virginia Aglietti, Xiaoyu Lu, Andrei Paleyes, and Javier Gonz´alez. Causal bayesian optimization. In International Conference on Artificial Intelligence and Statistics, pp. 3155–3164. PMLR, 2020.

[7] Virginia Aglietti, Neil Dhir, Javier Gonz´alez, and Theodoros Damoulas. Dynamic causal bayesian optimization. Advances in Neural Information Processing Systems, 34:10549–10560, 2021

[8] Virginia Aglietti, Alan Malek, Ira Ktena, and Silvia Chiappa. Constrained causal bayesian optimization. In International Conference on Machine Learning, pp. 304–321. PMLR, 2023.

[9] Limor Gultchin, Virginia Aglietti, Alexis Bellot, and Silvia Chiappa. Functional causal bayesian
optimization. In Uncertainty in Artificial Intelligence, pp. 756–765. PMLR, 2023.

[10] Scott Sussex, Anastasiia Makarova, and Andreas Krause. Model-based causal bayesian optimization. International Conference on Learning Representations, 2023.

[12] Nicola Branchini, Virginia Aglietti, Neil Dhir, and Theodoros Damoulas. Causal entropy optimization. In International Conference on Artificial Intelligence and Statistics, pp. 8586–8605. PMLR, 2023.

---

### Meta-Review · Area_Chair_uHaM · 2024-12-19

**Metareview:**

The submission makes the observation that for optimizing the value of a target variable (in a Bayesian framework but in general), it is sufficient to know/learn the causal parents (when there is no unobserved confounding affecting the target variable). They then propose a Bayesian approach for learning these parents and optimize the target objective under an additive (Gaussian) noise model in linear/nonlinear SCMs.

Two reviewers show strong and very strong support for the paper with scores of 8 and 10 (!), respectively and both with confidences of 4. One reviewer is strongly against and another is marginally against acceptance, especially after the rebuttal.

I agree with the critical expert reviewer that the novelty is limited because it is already well known that without latent confounders, it is sufficient to know/manipulate/optimize the parents in the causal graph. This was exploited in several existing causal bandit papers, e.g., see Lu et al. 2021 Causal Bandits with Unknown Graph Structure. Even the authors acknowledge this problem was solved by Lee and Bareinboim 2018. But the abstract present this known observation as one of the main contributions. The authors may not have intended this but current narrative diminishes the value of many existing results on this and needs to be changed. Given all this, I find it quite surprising that two reviewers gave an 8 and a 10 out of 10 and with confidence 4. My guess is that they are not very familiar with the existing causal bandit literature and they didn't pay close attention to other reviewers' criticism. Therefore, I will have to down-weigh their scores in making my decision.

I also recommend authors to cite such super-relevant papers in more depth. Currently, Lu et al is simply cited in a bulk of papers as "There has been work done on causal bandits with unknown graphs (Lu et al., 2021; Malek et al., 2023; Yan & Tajer, 2024), but the methods are restricted to specific graph types, additive functional relations or linear bandits". This is not doing justice to the prior literature in my opinion. I also recommend adding additive (Gaussian) noise in the title since the current title is very generic but the proposed method cannot handle arbitrary SCMs.

**Additional Comments On Reviewer Discussion:**

The authors seem to be not completely familiar with causal bandit literature, since during rebuttal period they said that Markov boundary is sufficient for optimization, which is not true as correctly pointed out by one of the reviewers. This was shown by Lee and Bareinboim via POMIS. The reviewers' initial opinion did not seem to have changed much after the rebuttal.

---

### Decision · Program_Chairs · 2025-01-22

Reject